# High-Fidelity Human Motion Generation with Motion Quality Feedback

## Abstract

Text-to-motion generation aims to synthesize realistic human motions from natural language descriptions. Prevailing approaches typically condition generative models on embeddings from the pre-trained CLIP text encoder. However, a fundamental discrepancy exists: CLIP's embeddings are optimized for static visual semantics, failing to capture the dynamic nuances essential for motion, consequently leading to suboptimal generation quality. To bridge this semantic gap, we propose AdaQF, a novel diffusion-based framework that enables the autonomous and efficient adaptation of the CLIP text encoder through feedback-driven co-optimization. AdaQF introduces a quality feedback loop, where semantic consistency constraints, between the generated motion, the conditioning text, and the ground truth motion, guide the fine-tuning of the CLIP encoder via low-rank adaptation. This process yields AdaCLIP, a motion-specialized text encoder that produces semantically rich and dynamic-aware embeddings. Our framework delivers advantages from three perspectives: it achieves state-of-the-art performance on standard benchmarks, achieving state-of-the-art results with an FID of 0.039 and an R-Precision of 0.888 on the HumanML3D database; it facilitates dramatically faster convergence (up to 8x); moreover, the resulting AdaCLIP module demonstrates remarkable transferability, serving as a versatile drop-in replacement that elevates the performance of various motion generation models including the VQ-VAE-based and latent diffusion-based ones, thus presenting a general and efficient solution for high-fidelity text-to-motion synthesis. The code of this paper will be released.

## 1 Introduction

Text-to-motion generation (Tevet et al., 2023; Zhang et al., 2024a; 2023a; Chen et al., 2023) seeks to synthesize realistic human motion sequences from natural language descriptions, with wide-ranging applications in animation (Hong et al., 2022), gaming (Holden et al., 2017), and robotics (Nishimura et al., 2020). Despite its vast potential, the task remains highly challenging because it requires not only an accurate understanding of complex textual instructions but also their faithful translation into high-quality, temporally coherent motion sequences.

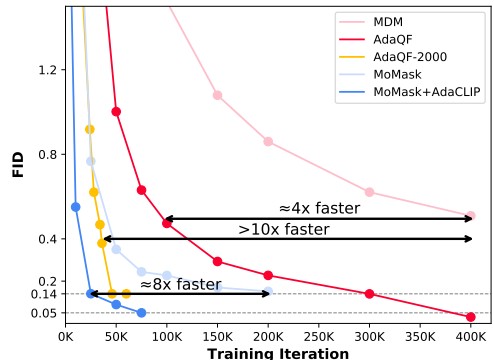

Figure 1: Our method remarkably improves the performance of two representative text-to-motion generation models (MDM and MoMask), even with only 2,000 HumanML3D samples (<10% of the training dataset). In addition to improving performance, our approach converges 4× to 8× faster than the vanilla models. The MoMask metric is evaluated using only the basic M-Transformer only.

Most existing text-to-motion generation methods, including the diffusion-based (Tevet et al., 2023; Zhang et al., 2024b; Huang et al., 2024a; Liang et al., 2024; Chen et al., 2023; Jin et al., 2024; Meng et al., 2025) and VQ-VAE-based (Zhang et al., 2023a; Pinyoanuntapong et al., 2023; Guo et al., 2024; Zou et al., 2024; Wan et al.,

2024; Huang et al., 2024b) methods, rely on a general text encoder (e.g., CLIP (Radford et al., 2021) in particular) to extract text embeddings from the conditional text. However, the training data for these general text encoders consists of image-text pairs or pure text only, which emphasizes object-level and static semantics, conflicting with the dynamic nature of motions required for motion generation and leading to weak adaptation. To address this issue, recent works (Lu et al., 2024; Maldonado et al., 2025; Li et al., 2025) propose pre-training a text encoder on text-motion pairs, typically employing contrastive learning to align its output embeddings with those of a motion encoder. This text encoder is then used to extract text embeddings for motion generation networks.

However, these methods (Lu et al., 2024; Maldonado et al., 2025) still suffer from several limitations. First, their text encoders (Lu et al., 2024; Maldonado et al., 2025) provide sub-optimal text embeddings for motion generation. This is mainly because contrastive training employs a manual training objective to pull paired text and motion embeddings closer than the unpaired ones, which does not necessarily align with the requirements of motion generation. In contrast, the ideal text embeddings for text-to-motion generation contain fine-grained dynamic cues to synthesize vivid and high-fidelity human motions. Second, they typically freeze the pre-trained text encoder, which prevents its co-optimization with the motion generator and reduces the overall optimization potential of the entire motion generation system (Bourigault & Bourigault, 2025).

Our key insight is that optimal text embeddings maximize synthesized-motion quality, rather than being selected according to manually defined embedding metrics. Accordingly, we propose **AdaQF**, a novel motion generation framework that autonomously **Ada**pts the CLIP text encoder to the motion generator, driven by motion **Q**uality **F**eedbacks. AdaQF consists of three main components: (i) an unfrozen CLIP text encoder that is to be finetuned via Low-Rank Adaptation (LoRA) (Hu et al., 2022), (ii) a diffusion-based motion generator, and (iii) a module that provides quality feedback from synthesized motions. In essence, we impose two high-level consistency constraints on the diffusion model's outputs during training, which act as semantic feedback signals to update the entire AdaQF system. First, we extract the embeddings for the conditional text, predicted motion, and ground-truth motion, respectively. The embedding extractors are trained with a contrastive objective to construct a unified embedding space where positive text–motion pairs are closely aligned. We then introduce two semantic consistency constraints to provide semantic feedback for training AdaQF: a motion consistency constraint is to measure the similarity between the predicted and ground-truth motions, and a text consistency constraint is to measure the similarity of the predicted motions and their conditional texts. These constraints guide the CLIP text encoder to produce text embeddings that better support high-fidelity motion generation. The above approaches yield our adapted CLIP text encoder named AdaCLIP, which extracts motion-aware embeddings and boosts the quality of generated motions. During inference, the quality feedback is no longer required, so AdaQF introduces no additional inference cost compared to the vanilla diffusion model.

We emphasize that the prior learned by AdaCLIP is more transferable than the vanilla CLIP text encoder. In practice, AdaCLIP acts as a plug-and-play replacement for the original CLIP text encoder in diverse motion generation paradigms. Specifically, we deploy AdaCLIP to VQ-VAE-based models (e.g., MoMask (Guo et al., 2024)) and latent-diffusion-based models (e.g., SALAD (Hong et al., 2025)). We keep all the other training settings unchanged and retrain them. This straightforward substitution consistently produces substantial performance improvements, which highlights AdaCLIP's transferability.

Beyond the transferability, AdaQF also brings substantial gains in training efficiency. It converges roughly 4× faster than our baseline MDM owing to the mutual adaptation between AdaCLIP and the diffusion model, as is illustrated in Figure 1. Moreover, AdaQF exhibits strong learning capability under limited training data, achieving competitive FID scores with over 10× acceleration in convergence. The fast convergence makes it practical to be trained on very large-scale datasets. Equipped with our AdaCLIP, other motion generation models such as MoMask (Guo et al., 2024) can converge much more efficiently, achieving up to approximately 8× faster convergence than their vanilla counterparts. These results highlight the effectiveness and potential of our approach as a general solution for enhancing text-to-motion generation.

Our main contributions can be summarized as follows:

- We emphasize the importance of enabling the text encoder to autonomously learn optimal text embeddings, which maximize synthesized-motion quality.

- We introduce AdaQF, a novel motion generation framework that autonomously adapts the CLIP text encoder to the motion generator, driven by motion quality feedback.
- Our AdaQF achieves state-of-the-art results on the HumanML3D and KIT-ML benchmarks in terms of FID and R-Precision. The resulting AdaCLIP exhibits strong transferability across different generative frameworks and enables substantially faster convergence, achieving up to an 8× speedup.
- Our AdaQF demonstrates strong data efficiency, achieving the best overall performance even with limited training data.

## 2 RELATED WORKS

**Text-Conditioned Human Motion Generation.** Mainstream text-to-motion generation methods can be primarily categorized into vector-quantized variational autoencoders (VQVAE) and diffusion-based methods. VQVAE-based methods (Guo et al., 2022; Zhang et al., 2023a; Jiang et al., 2024; Guo et al., 2024; Pinyoanuntapong et al., 2023) adopt a token prediction network to map the text condition into the categorical distribution of tokens. Then they map the predicted tokens into a motion sequence via a motion decoder. According to the mode of token prediction, existing approaches can be further classified into two sub-categories: (1) autoregressive prediction methods (Zhang et al., 2023a; Zhong et al., 2023) that predict the next token autoregressively (2) masked modeling methods (Guo et al., 2024; Pinyoanuntapong et al., 2023; Zhang et al., 2025) that predict the next set of tokens with bilateral attention. T2M-GPT (Zhang et al., 2023a) generates motion sequences based on the textual condition and the history motion tokens via a transformer. MoMask (Guo et al., 2024) employs a residual VQ-VAE with multi-level codebooks coupled with the masked modeling generative paradigmand further enhances the generation qualities. Diffusion-based methods (Ho et al., 2020) model human motion representations by denoising the noisy input step-by-step. Based on the representation space where the diffusion process occurs, diffusion-based methods can be categorized into standard (Tevet et al., 2023; Zhang et al., 2024a; Kim et al., 2023) and latent diffusion-based (Chen et al., 2023; Jin et al., 2024; Hong et al., 2025) strategies.

The most related to ours is MoCLIP (Maldonado et al., 2025) and LAMP (Li et al., 2025). MoCLIP (Maldonado et al., 2025) finetunes the vanilla CLIP text encoder via contrastive learning and replaces the finetuned CLIP text encoder with the vanilla one. LAMP (Li et al., 2025) trains a pair of text encoder and motion encoder using multi-task pretraining. However, as analyzed in Section 1, this strategy may produce sub-optimal text embeddings for motion generation models. In comparison, we co-optimize the CLIP text encoder with the motion generator and enable autonomous adaptation of the CLIP text encoder according to quality feedback on synthesized motions.

## 3 METHOD

The overview of our AdaQF is illustrated in Figure 2. Given a text prompt $c$ as a condition, our goal is to generate a motion sequence $x \in \mathbb{R}^{N \times D}$, where $N$ is the length of the motion sequence and $D$ is the dimension of human pose representations. Our AdaQF consists of three main components: (i) two extractors for getting motion embeddings and text embeddings, (ii) semantic consistency constraints that provide the quality feedback from the synthesized motions, and (iii) we use LoRA to add a learnable branch for the vanilla CLIP text encoder. We first briefly introduce the preliminaries of the diffusion model in Section 3.1. We then elaborate on the module that enables motion quality feedback in Section 3.2. Next, we explain the way to deploy motion quality feedback during motion generator training in Section 3.3. Finally, we use LoRA to add learnable branches to the vanilla CLIP, allowing CLIP to learn optimal text embeddings and co-optimizes it with the main diffusion model in Section 3.4.

### 3.1 PRELIMINARIES

*Diffusion process.* Starting with a motion sequence $x_0$ sampled from the training data distribution, the diffusion process transforms $x_0$ into noisy data $\{x_t\}_{t=1}^{T}$ with varying noise levels, where $T$ is the maximum diffusion process step. This process is formulated as $x_t = \sqrt{1 - \beta_t} x_{t-1} + \sqrt{\beta_t} \epsilon$, where $\beta_t$ is the noise scheduler and $\epsilon$ is gaussian noise. A notable property of the forward process is that it

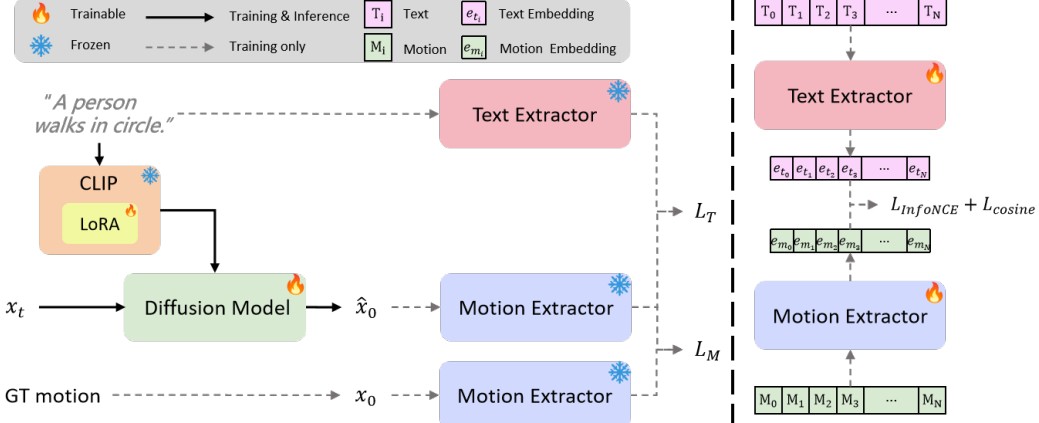

Figure 2: **Framework overview.** (Left) Our framework is composed of an unfrozen CLIP text encoder named AdaCLIP, a diffusion-based motion generator, and a module that provides motion quality feedback. The overall framework is optimized by the basic mean square error (MSE) loss, the text consistency constraint $L_T$, and the motion consistency constraint $L_M$. During inference, only AdaCLIP and the diffusion-based motion generator remain. (Right) Before training AdaQF, we train the pair of text and motion extractors in advance using an InfoNCE loss and a cosine similarity loss.

admits sampling $x_t$ at an arbitrary timestep $t$ in one step:

$$x_t = \sqrt{1 - \beta_t} x_0 + \sqrt{\beta_t} \epsilon = \boldsymbol{\alpha}_t x_0 + \boldsymbol{\sigma}_t \epsilon. \tag{1}$$

*Denoising process.* The denoising process models the conditional distribution $p_\theta(\hat{x}_0 | x_T, c)$ based on the text prompt $c$ and obtains the synthesized motion $\hat{x}_0$. This was achieved using a denoising network to denoise $x_t$ to $\hat{x}_0$, changing $t$ from $T$ to 1.

### 3.2 TRAINING TEXT AND MOTION EXTRACTORS

The text extractor $E_T$ and motion extractor $E_M$ aim to map the conditional text $c$, the predicted motion $\hat{x}_0$, and the ground-truth motion $x_0$ into corresponding embeddings $e_c$, $\hat{e}_m$, and $e_m$. This embedding space serves as a criterion for text-motion pairs with two key properties: (1) the text and motion embeddings of positive samples in this space should be highly aligned, and (2) embeddings within a mini-batch should remain mutually highly discriminative.

Specifically, we build the text and motion extractors using Gated Recurrent Unit (GRU) following T2M (Guo et al., 2022). The text extractor takes the conditional text and its length as input, and employs a GRU layer to extract sequential text features, with the final hidden state serving as the resulting text embedding. Similarly, the motion extractor takes the motion sequence and its length as input, and outputs the last hidden state of the GRU as the motion embedding.

We adopt both the InfoNCE loss (Oord et al., 2018) and the cosine similarity loss to train these extractors. The InfoNCE loss encourages cross-modal alignment by pulling paired text–motion samples closer while pushing apart unpaired ones within a batch. In parallel, the cosine similarity loss is imposed on paired text-motion samples and explicitly maximizes their angular similarity, reinforcing fine-grained alignment of semantically corresponding samples.

Given a batch of $N$ paired text-motion embeddings $(e_{t_1}, e_{m_1}), \cdots, (e_{t_N}, e_{m_N})$, we define the pairs with $i = j$ as positive samples and those $(e_{t_i}, e_{m_j})$ with $i \neq j$ as negative samples. We obtain the similarity matrix by computing the pairwise cosine similarities for all pairs of embeddings in the batch, where each element in this matrix is denoted as $S_{ij} = cos(e_{t_i}, e_{m_j})$. The loss function to train the text and motion extractors can be formulated as follows:

$$\mathcal{L}_{\text{extracors}} = -\frac{1}{2N} \sum_i \left( -\log \frac{\exp(S_{ii}/\tau)}{\sum_j \exp(S_{ij}/\tau)} - \log \frac{\exp(S_{ii}/\tau)}{\sum_j \exp(S_{ji}/\tau)} \right) + \frac{1}{N} \sum_i S_{ii}. \tag{2}$$

Combining these two objectives enables the extractors to learn embeddings that are both discriminative across negative pairs and highly consistent across positive pairs, which provides a rich and motion-specific prior space for the subsequent training of AdaQF.

### 3.3 MODEL TRAINING WITH QUALITY FEEDBACKS

Most existing text-to-motion generation methods (Tevet et al., 2023; Chen et al., 2023; Xie et al., 2024; Zhang et al., 2023a; Guo et al., 2024) rely primarily on the mean squared error (MSE) loss as the training objective. While effective for element-wise reconstruction, MSE loss provides only low-level supervision. This often biases the model toward memorizing local joint trajectories frame by frame, while neglecting higher-level semantic patterns in motion sequences that are essential for motion quality. For example, when learning a walking motion, minimizing MSE encourages the network to reproduce the exact positions of each joint at every frame, but a walking motion should still be recognized as such even with slight variations in body joints.

To address this limitation, we propose two complementary consistency constraints: Motion Consistency Constraint and Text Consistency Constraint, which are denoted as $\mathcal{L}_M$ and $\mathcal{L}_T$, respectively. The former enforces in-modal semantic consistency between generated and ground-truth motions, capturing structural and perceptual characteristics beyond frame-wise alignment. The latter ensures cross-modal semantic consistency by constraining the generated motion to remain consistent with its textual description. These two consistency constraints regularize the training of AdaQF, supplementing the insufficient semantic constraint capabilities of the original MSE loss.

*Motion Consistency Constraint* is designed as a cosine similarity loss in the embedding space during training. Let $x_0$ denote the ground-truth motion sequence and $\hat{x}_0$ the generated motion produced by the diffusion model. Both sequences are projected into an embedding space by the motion extractor $E_M$: $\hat{e_m} = E_M(\hat{x}_0)$ and $e_m = E_M(x_0)$. Thus, the motion consistency constraint $\mathcal{L}_M(\hat{e_m}, e_m)$ is formulated as follows:

$$\mathcal{L}_M(\hat{e_m}, e_m) = 1 - cos(\hat{e_m}, e_m) = 1 - \frac{\hat{e_m} \cdot e_m}{\|\hat{e_m}\|\|e_m\|}. \tag{3}$$

*Text Consistency Constraint* is also designed as a cosine similarity loss in the embedding space, providing cross-modal semantic supervision. We embed both the generated motion $\hat{x}_0$ and the input textual description $c$ into the embedding space using the text extractor $E_T$ and the motion extractor $E_M$, respectively: $\hat{e_m} = E_M(\hat{x}_0)$ and $e_t = E_M(c)$. Thus, the text consistency constraint $\mathcal{L}_T(\hat{e_m}, e_t)$ is formulated as follows:

$$\mathcal{L}_T(\hat{e_m}, e_t) = 1 - cos(\hat{e_m}, e_t) = 1 - \frac{\hat{e_m} \cdot e_t}{\|\hat{e_m}\|\|e_t\|}. \tag{4}$$

### 3.4 FINETUNING THE PRE-TRAINED CLIP TEXT ENCODER

Having trained the text and motion extractors in Section 3.2, our next goal is to jointly optimize CLIP and the motion diffusion model. Due to the limited size of text-to-motion datasets (Guo et al., 2022; Plappert et al., 2016), the key challenge is how to incorporate motion-specific knowledge into the pre-trained CLIP, meanwhile maintaining its general text understanding ability. To address this, we use the Low-Rank Adaptation (LoRA) technique (Hu et al., 2022), a parameter-efficient fine-tuning method that inserts learnable low-rank matrices into existing weight matrices while keeping the majority of pre-trained parameters frozen, to add learnable parameters for the vanilla CLIP. This allows the model to absorb new motion-specific priors without compromising its general priors. In practice, we apply LoRA to the projection matrices of Q, K, and V in Multi-Head Attention, as well as the MLP layers, using a uniform rank across all modules for simplicity. We demonstrate that the CLIP enhanced by LoRA is well-adapted to motion generation; thus, we refer to it as AdaCLIP throughout the rest of the paper.

Combining the consistency constraints in 3.3, feedback signals are back-propagated through the motion diffusion model into AdaCLIP, implicitly guiding it to integrate motion-specific priors. Through this co-optimization, AdaCLIP and the motion diffusion model converge to a mutually adaptive state.

After training, we obtain an AdaCLIP that is not only aligned with motion semantics but also well-suited for text-to-motion generation. We further demonstrate that this AdaCLIP can be used in place of the original CLIP in other types of motion generation frameworks in Section 4.2.

**Training Objectives.** The overall training objective of our AdaQF can be formulated as follows:

$$\mathcal{L} = \mathcal{L}_{mse} + \lambda_M \mathcal{L}_M + \lambda_T \mathcal{L}_T, \tag{5}$$

where $\mathcal{L}_{mse}$ denotes the MSE loss between the predicted motion and the ground-truth motion. $\lambda_M$ and $\lambda_T$ are hyperparameters to the strength of $\mathcal{L}_M$ and $\mathcal{L}_T$.

## 4 EXPERIMENTS

We conduct comprehensive experiments on two mainstream text-to-motion datasets, HumanML3D (Guo et al., 2022) and KIT-ML (Plappert et al., 2016). HumanML3D uses motion sequences from the AMASS (Mahmood et al., 2019) and HumanAct12 (Guo et al., 2020) collections, re-annotates them. It contains 14,616 motion sequences annotated with 44,970 textual descriptions, comprising 5,371 unique words. The total duration sums up to 28.59 hours. KIT-ML includes 3,911 motion sequences and 6,353 sequence-level natural language descriptions. In our experiments, we use motion sequences with a maximum length of 196 frames. HumanML3D is a large-scale dataset, whereas KIT-ML is relatively small. This allows us to evaluate our methods in both resource-rich and resource-limited settings.

**Evaluation Metrics.** We adopt the metrcis proposed by T2M (Guo et al., 2022) to evaluate the quantitative performance of our method: (1) *Fréchet Inception Distance (FID)* evaluates the quality of generated motions by measuring the distributional difference between the generated motions and real motions. (2) *R-Precision* and *Multimodal Distance* evaluate the semantic alignment between input text and generated motions. *R-Precision* is the motion-retrieval precision between texts and generated motions, and the *Multimodal Distance* calculates the distance between motions and texts. (3) *Diversity* indicates the variance in generated motions, and *MultiModality*, which assesses the diversity of motions generated from the same text description, are also adopted as secondary metrics. They are considered less important than the generation quality and alignment with the input text. We emphasize that FID and R-precision are the most important metrics, while Diversity and Multi-Modality are the secondary metrics that should be assessed in conjunction with primary performance metrics such as FID and R-Precision (Guo et al., 2024; Hong et al., 2025).

### 4.1 IMPLEMENTATION DETAILS

We select MDM (Tevet et al., 2023), equipped with a Transformer decoder backbone and a CLIP text encoder, as the baseline of AdaQF. The text embeddings output by CLIP are injected via cross-attention. The maximum diffusion step is set to 50 during both training and inference. The training process is executed on a single NVIDIA 4090 GPU. On the HumanML3D benchmark, we use the initial learn rate 1e-4 and degrade by 0.1 at 300,000 iterations, and the max training iteration is set to 600,000. The training batch size is set to 64. On the KIT-ML benchmark, we use the same learning rate schedule and max training iteration as HumanML3D. The training batch size is set to 32. We also use classifier-free guidance (CFG) (Ho & Salimans, 2022) for a better trade-off between motion quality and text alignment degree. We set the CFG weight to 2.5 for both HumanML3D and KIT-ML benchmarks following the MDM's setting. For finetuning the CLIP text encoder, the rank value of LoRA is set to 16 across all modules without extensive hyperparameter tuning.

### 4.2 COMPARISON TO STATE-OF-THE-ART APPROACHES

We test our framework using the same protocol as in (Guo et al., 2022; Tevet et al., 2023). The reported metric values represent the mean with a 95% statistical confidence interval. We perform the quantitative comparison with previous methods on the HumanML3D (Guo et al., 2022) and KIT-ML (Plappert et al., 2016) benchmarks, and the results are shown in Table 1 and 2.

Our method surpasses previous approaches and achieves state-of-the-art performance on FID and R-precision metrics across both HumanML3D and KIT-ML benchmarks. This emphasizes the significance of enabling the text encoder to learn optimal text embeddings based on the quality feedback

| | Methods | R-Precision↑ | | | FID↓ | MultiModal Dist↓ | MultiModality↑ |
|---|---|---|---|---|---|---|---|
| | | Top 1 | Top 2 | Top 3 | | | |
| VAE/VQ | T2M (Guo et al., 2022) | $0.455^{\pm.003}$ | $0.636^{\pm.003}$ | $0.736^{\pm.002}$ | $1.087^{\pm.021}$ | $3.347^{\pm.008}$ | $2.219^{\pm.074}$ |
| | T2M-GPT (Zhang et al., 2023a) | $0.492^{\pm.003}$ | $0.679^{\pm.002}$ | $0.775^{\pm.002}$ | $0.141^{\pm.005}$ | $3.121^{\pm.009}$ | $1.831^{\pm.048}$ |
| | MMM (Pinyoanuntapong et al., 2023) | $0.504^{\pm.003}$ | $0.696^{\pm.002}$ | $0.794^{\pm.002}$ | $0.080^{\pm.003}$ | $2.998^{\pm.007}$ | $1.164^{\pm.041}$ |
| | MoMask (Guo et al., 2024) | $0.521^{\pm.002}$ | $0.713^{\pm.002}$ | $0.807^{\pm.002}$ | $0.045^{\pm.002}$ | $2.958^{\pm.008}$ | $1.241^{\pm.040}$ |
| | BAMM (Pinyoanuntapong et al., 2024) | $0.525^{\pm.002}$ | $0.720^{\pm.003}$ | $0.814^{\pm.003}$ | $0.055^{\pm.002}$ | $2.919^{\pm.008}$ | $1.687^{\pm.051}$ |
| | KinMo (Zhang et al., 2025) | $0.529^{\pm.003}$ | $0.722^{\pm.002}$ | $0.817^{\pm.002}$ | $0.050^{\pm.003}$ | $2.907^{\pm.009}$ | $1.313^{\pm.041}$ |
| | MoMask+MoCLIP (Maldonado et al., 2025) | $0.533^{\pm.003}$ | $\underline{0.730}^{\pm.002}$ | $\underline{0.823}^{\pm.001}$ | $0.047^{\pm.002}$ | $\underline{2.868}^{\pm.006}$ | $1.242^{\pm.040}$ |
| | HGM³ (Jeong et al., 2025) | $\underline{0.535}^{\pm.002}$ | $0.726^{\pm.002}$ | $0.822^{\pm.002}$ | $\underline{0.036}^{\pm.002}$ | $2.904^{\pm.008}$ | $1.206^{\pm.051}$ |
| | MoMask+DisCoRD(Cho et al., 2025) | $0.524^{\pm.003}$ | $0.715^{\pm.003}$ | $0.809^{\pm.002}$ | $\mathbf{0.032}^{\pm.002}$ | $2.938^{\pm.010}$ | $\underline{1.288}^{\pm.043}$ |
| | LAMP (Li et al., 2025) | $\mathbf{0.557}^{\pm.003}$ | $\mathbf{0.751}^{\pm.002}$ | $\mathbf{0.843}^{\pm.001}$ | $\mathbf{0.032}^{\pm.002}$ | $2.759^{\pm.007}$ | - |
| Diffusion | MotionDiffuse (Zhang et al., 2024a) | $0.491^{\pm.001}$ | $0.681^{\pm.001}$ | $0.782^{\pm.001}$ | $0.630^{\pm.001}$ | $3.113^{\pm.001}$ | $1.553^{\pm.042}$ |
| | MDM (Tevet et al., 2023) | $0.455^{\pm.006}$ | $0.645^{\pm.004}$ | $0.749^{\pm.002}$ | $0.489^{\pm.047}$ | $3.330^{\pm.025}$ | $2.290^{\pm.070}$ |
| | MLD (Chen et al., 2023) | $0.481^{\pm.003}$ | $0.673^{\pm.003}$ | $0.772^{\pm.002}$ | $0.473^{\pm.013}$ | $3.196^{\pm.010}$ | $2.413^{\pm.079}$ |
| | ReMoDiffuse (Zhang et al., 2023b) | $0.510^{\pm.005}$ | $0.698^{\pm.006}$ | $0.795^{\pm.004}$ | $0.103^{\pm.004}$ | $2.974^{\pm.016}$ | $1.795^{\pm.043}$ |
| | StableMoFusion (Huang et al., 2024a) | $0.553^{\pm.003}$ | $0.748^{\pm.002}$ | $0.841^{\pm.002}$ | $0.098^{\pm.003}$ | - | $1.774^{\pm.051}$ |
| | SALAD (Hong et al., 2025) | $0.581^{\pm.003}$ | $\underline{0.769}^{\pm.003}$ | $\underline{0.857}^{\pm.002}$ | $0.076^{\pm.002}$ | $2.649^{\pm.009}$ | $1.751^{\pm.062}$ |
| | **AdaQF(ours)** | $\mathbf{0.627}^{\pm.004}$ | $\mathbf{0.811}^{\pm.003}$ | $\mathbf{0.888}^{\pm.002}$ | $0.039^{\pm.001}$ | $\mathbf{2.382}^{\pm.010}$ | $1.184^{\pm.042}$ |

Table 1: **Quantitative results on the HumanML3D test set.** We separate the methods into diffusion-based methods and VQ-based methods. ± indicates a 95% confidence interval. **Bold** face indicates the best result, while underscore refers to the second best.

| | Methods | R Precision↑ | | | FID↓ | MultiModal Dist↓ | MultiModality↑ |
|---|---|---|---|---|---|---|---|
| | | Top 1 | Top 2 | Top 3 | | | |
| VAE/VQ | T2M (Guo et al., 2022) | $0.361^{\pm.005}$ | $0.559^{\pm.007}$ | $0.681^{\pm.007}$ | $3.022^{\pm.107}$ | $3.488^{\pm028}$ | $2.052^{\pm.107}$ |
| | T2M-GPT (Zhang et al., 2023a) | $0.416^{\pm.006}$ | $0.627^{\pm.006}$ | $0.745^{\pm.006}$ | $0.514^{\pm.029}$ | $3.007^{\pm.023}$ | $1.570^{\pm.039}$ |
| | MMM (Pinyoanuntapong et al., 2023) | $0.404^{\pm.005}$ | $0.621^{\pm.004}$ | $0.744^{\pm.005}$ | $0.316^{\pm.028}$ | $2.977^{\pm.019}$ | $1.232^{\pm.032}$ |
| | MoMask (Guo et al., 2024) | $0.433^{\pm.007}$ | $0.656^{\pm.005}$ | $0.781^{\pm.005}$ | $0.204^{\pm.011}$ | $2.779^{\pm.022}$ | $1.131^{\pm.043}$ |
| | BAMM (Pinyoanuntapong et al., 2024) | $0.438^{\pm.009}$ | $0.661^{\pm.009}$ | $0.788^{\pm.005}$ | $0.183^{\pm.013}$ | $2.723^{\pm.026}$ | $1.609^{\pm.065}$ |
| | HGM³ (Jeong et al., 2025) | $\underline{0.444}^{\pm.007}$ | $\underline{0.664}^{\pm.005}$ | $\underline{0.791}^{\pm.006}$ | $0.176^{\pm.010}$ | $\underline{2.710}^{\pm.019}$ | $1.152^{\pm.041}$ |
| | MoMask+DisCoRD (Cho et al., 2025) | $0.434^{\pm.007}$ | $0.657^{\pm.005}$ | $0.775^{\pm.004}$ | $\underline{0.169}^{\pm.010}$ | $2.792^{\pm.015}$ | $\underline{1.266}^{\pm.046}$ |
| | LAMP (Li et al., 2025) | $\mathbf{0.479}^{\pm.006}$ | $\mathbf{0.691}^{\pm.005}$ | $\mathbf{0.826}^{\pm.005}$ | $\mathbf{0.141}^{\pm.013}$ | $\mathbf{2.704}^{\pm.018}$ | - |
| Diffusion | MotionDiffuse (Zhang et al., 2024a) | $0.417^{\pm.004}$ | $0.621^{\pm.004}$ | $0.739^{\pm.004}$ | $1.954^{\pm.062}$ | $2.958^{\pm.005}$ | $0.730^{\pm.013}$ |
| | MDM (Tevet et al., 2023) | $0.471^{\pm.004}$ | $0.657^{\pm.004}$ | $0.765^{\pm.004}$ | $0.322^{\pm.018}$ | $2.802^{\pm.022}$ | - |
| | MLD (Chen et al., 2023) | $0.390^{\pm.008}$ | $0.609^{\pm.008}$ | $0.734^{\pm.007}$ | $0.404^{\pm.027}$ | $3.204^{\pm.027}$ | $2.192^{\pm.071}$ |
| | ReMoDiffuse (Zhang et al., 2023b) | $0.427^{\pm.014}$ | $0.641^{\pm.004}$ | $0.765^{\pm.055}$ | $\mathbf{0.155}^{\pm.006}$ | $2.814^{\pm.012}$ | $1.239^{\pm.028}$ |
| | StableMoFusion (Huang et al., 2024a) | $0.445^{\pm.006}$ | $0.660^{\pm.005}$ | $0.782^{\pm.004}$ | $0.258^{\pm.029}$ | - | $1.362^{\pm.062}$ |
| | SALAD (Hong et al., 2025) | $\underline{0.477}^{\pm.006}$ | $\underline{0.711}^{\pm.005}$ | $\underline{0.828}^{\pm.005}$ | $0.296^{\pm.017}$ | $\underline{2.585}^{\pm.016}$ | $1.004^{\pm.040}$ |
| | **AdaQF(ours)** | $\mathbf{0.494}^{\pm.005}$ | $\mathbf{0.727}^{\pm.005}$ | $\mathbf{0.846}^{\pm.005}$ | $\underline{0.161}^{\pm.010}$ | $\mathbf{2.513}^{\pm.018}$ | $1.631^{\pm.062}$ |

Table 2: **Quantitative evaluation on the KIT-ML test set.** We separate the methods into diffusion-based methods and VQ-based methods. ± indicates a 95% confidence interval. **Bold** face indicates the best result, while underscore refers to the second best.

of generated motions, further validating our analysis in Section 1 and demonstrating the superiority of our approach. Notably, compared to our baseline MDM, the FID improves from 0.489 to 0.039, an improvement of over 90%, and the R-precision increases from 0.749 to 0.888, showcasing the effectiveness of the feedback-driven co-optimization.

**Qualitative Comparisons.** Figure 3 displays qualitative comparisons of our AdaQF and three representative methods MDM (Tevet et al., 2023), MoMask (Guo et al., 2024), SALAD (Hong et al., 2025). MDM typically understands the text instruction roughly, fails to execute some details like *"dodge something"*. MoMask and SALAD could play relatively fluent motions; nevertheless, they may still lose some details like *"sitting on the floor"* and *"jumping forward"*. For the understanding of complex text instruction of the second sample in Figure 3, precisely execute each detail like *"bring both hands to his face"*. For the fourth example, AdaQF faithfully and precisely converts the text *"counter-clockwise"* and *"left arm out"* to realistic waltz motion. For more visualization results, please see supplementary videos for clearer visuals.

## 4.3 Transferability of AdaCLIP across Generation Models

Our AdaCLIP has been injected with motion-specific priors that can be transferred to other motion generation frameworks. To demonstrate this prior transferability, we substitute AdaCLIP trained on the HumanML3D dataset for the vanilla CLIP in two state-of-the-art models from distinct generative paradigms: MoMask (Guo et al., 2024) (a VQ-VAE-based framework) and SALAD (Hong et al., 2025) (a latent-diffusion-based framework), keeping all other training settings unchanged and retrain them on the HumanML3D dataset. We evaluate their performance and report the results in

| Methods | R Precision↑ | | | FID↓ | MultiModal Dist↓ | MultiModality↑ |
|---|---|---|---|---|---|---|
| | Top 1 | Top 2 | Top 3 | | | |
| MDM (Tevet et al., 2023) | $0.455^{\pm.006}$ | $0.645^{\pm.007}$ | $0.749^{\pm.006}$ | $0.489^{\pm.047}$ | $3.330^{\pm.025}$ | $2.290^{\pm.070}$ |
| MoMask (Guo et al., 2024) | $0.521^{\pm.002}$ | $0.713^{\pm.002}$ | $0.807^{\pm.002}$ | $0.045^{\pm.002}$ | $2.958^{\pm.008}$ | $1.241^{\pm.040}$ |
| MoMask+MoCLIP (Maldonado et al., 2025) | $0.533^{\pm.003}$ | $0.730^{\pm.002}$ | $0.823^{\pm.001}$ | $0.047^{\pm.002}$ | $2.868^{\pm.006}$ | $1.242^{\pm.040}$ |
| SALAD (Hong et al., 2025) | $0.581^{\pm.003}$ | $0.769^{\pm.003}$ | $0.857^{\pm.002}$ | $0.076^{\pm.002}$ | $2.649^{\pm.009}$ | $1.751^{\pm.062}$ |
| **MDM+AdaCLIP(ours)** | $0.573^{\pm.003}$ | $0.752^{\pm.004}$ | $0.838^{\pm.005}$ | $0.149^{\pm.005}$ | $2.737^{\pm.013}$ | $1.083^{\pm.052}$ |
| **MoMask+AdaCLIP(ours)** | $0.554^{\pm.003}$ | $0.751^{\pm.002}$ | $0.842^{\pm.003}$ | $\mathbf{0.040}^{\pm.002}$ | $2.771^{\pm.009}$ | $1.165^{\pm.044}$ |
| **SALAD+AdaCLIP(ours)** | $\mathbf{0.590}^{\pm.003}$ | $\mathbf{0.779}^{\pm.002}$ | $\mathbf{0.864}^{\pm.001}$ | $0.061^{\pm.003}$ | $\mathbf{2.625}^{\pm.007}$ | $1.400^{\pm.056}$ |

Table 3: **Transferability experiments on HumanML3D dataset.** We test the transferability of AdaCLIP across different generative frameworks.

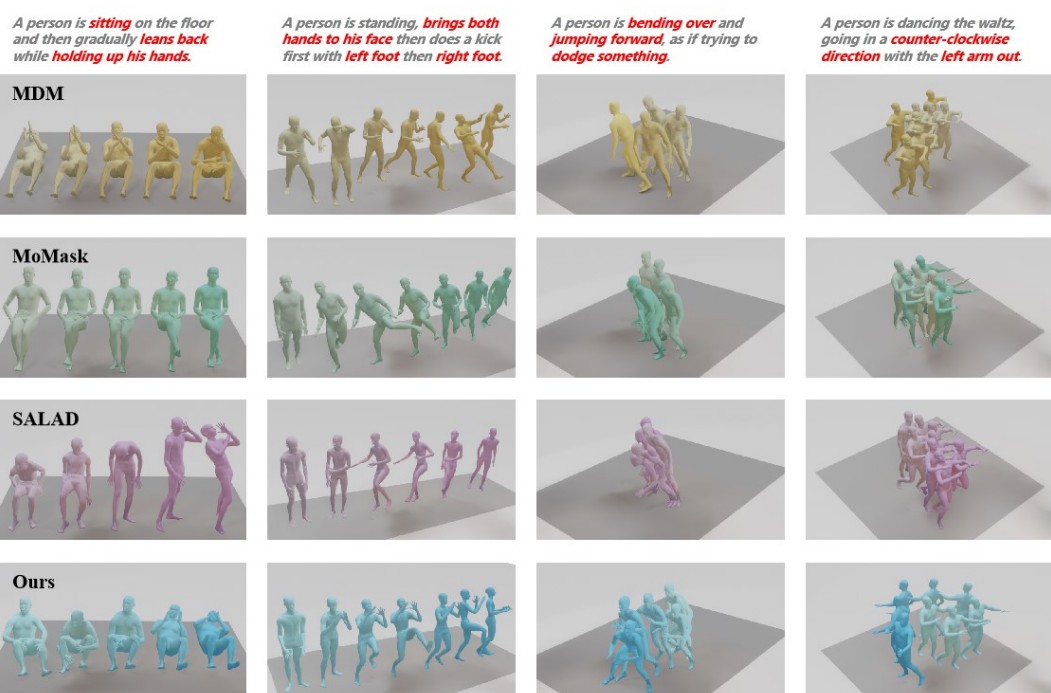

Figure 3: Qualitative comparison of results generated by various methods, including MDM (Tevet et al., 2023), MoMask (Guo et al., 2024), SALAD (Hong et al., 2025) and our approach. The key actions are highlighted in red.

Table 3. Both models consistently show clear performance improvements on FIDs and R-Precision and achieve the state-of-the-art results, demonstrating the strong transferability of AdaCLIP. When comparing with our most related work, MoCLIP (Maldonado et al., 2025), which is used in a plug-and-play manner, MoMask+AdaCLIP still consistently performs better than MoMask+MoCLIP on FIDs and R-Precision.

## 4.4 ABLATION STUDIES

**Ablations on proposed components.** We conduct several ablation experiments to evaluate the contribution of each proposed component. Specifically, the ablations include: 1) MDM, the baseline, 2) applying consistency constraints on MDM, and 3) applying consistency constraints on MDM while co-optimizing the CLIP text encoder. Here we introduce another metric named Skating Ratio to evaluate the physical fidelity, which is a proxy for the physical plausibility of motions, following (Karunratanakul et al., 2023) and (Xie et al., 2024). It measures the proportion of frames in which either foot skates more than a certain distance (2.5 cm) while maintaining contact with the ground (foot height < 5 cm). We present the quantitative results in Table 4. For a more detailed analysis of the evolution of the metrics of FID, R-Precision, Skating Ratio, and MSE loss during training,

| Ablation | Method | Consistency Constraints | Tuning CLIP | FID $\downarrow$ | **R-precision** $\uparrow$ (Top-3) | Skating $\uparrow$ Ratio |
|---|---|---|---|---|---|---|
| ① | MDM (baseline) | | | $0.489^{\pm .047}$ | $0.749^{\pm .006}$ | $0.095^{\pm .0009}$ |
| ② | with consistency constraints | ✓ | | $0.098^{\pm .002}$ | $0.838^{\pm .002}$ | $0.069^{\pm .0008}$ |
| ③ | with LoRA tuning | | LoRA | $0.566^{\pm .004}$ | $0.802^{\pm .002}$ | $0.107^{\pm .0007}$ |
| ④ | **Ours (full model)** | ✓ | LoRA | $\mathbf{0.039}^{\pm .001}$ | $\mathbf{0.888}^{\pm .002}$ | $\mathbf{0.061}^{\pm .0007}$ |
| ⑤ | Ours with full tuning | ✓ | Full | $0.073^{\pm .002}$ | $0.873^{\pm .002}$ | $0.063^{\pm .0007}$ |

Table 4: Ablation studies of proposed components on HumanML3D test set.

please refer to APPENDIX A.5. As shown in Table 4, Ablation ① is our baseline MDM. Ablation ② reveals that even without finetuning the CLIP text encoder, the consistency constraints still regularize the motion diffusion model to produce better motions and perform competitive quantitative results. Ablation ③ shows that solely finetuning the CLIP without using the consistency constraints substantially deteriorates the generation quality. We attribute this degradation to shortcut learning. When the CLIP text encoder is unfrozen but trained solely under the low-level MSE loss, the system of CLIP and diffusion is implicitly optimized to minimize reconstruction error rather than learning the text-to-motion mapping. As a consequence, the CLIP encoder gradually deviates from producing meaningful text embeddings, while the diffusion model simultaneously adapts to interpret these distorted embeddings. This leads to text embeddings that overfit to the training distribution and fail to generalize to unseen descriptions, ultimately harming generation quality. In contrast, in Ablation ④, combining the consistency constraints with CLIP tuning yields significant improvement. This indicates that the semantic consistency constraints indeed regularize the diffusion model's training and prevent it from capturing only low-level patterns. They also highlight the importance of allowing the text encoder to learn optimal text embeddings for text-to-motion generation adaptively.

**LoRA-based finetuning or Full-parameter finetuning.** We chose to use LORA for fine-tuning based on the following two reasons. First, we hope to preserve the pretrained prior of the original CLIP text encoder while adapting to the text-to-motion task. Second, the HumanML3D and KIT-ML datasets are much smaller than those used for training CLIP, and using LORA can significantly reduce the risk of overfitting. We conduct experiments on finetuning CLIP with full-parameter tuning, as reported in Ablation ⑤ of Table 4. Full-parameter finetuning yields worse generation quality than our default LoRA-based tuning, by a clear margin. This demonstrates that unrestricted updates tend to destabilize CLIP's pretrained prior and cause overfitting, whereas LoRA provides a more stable and effective way to specialize the text encoder for motion generation.

**Abaltions on $\lambda_M$ and $\lambda_T$.** Our AdaQF only involves two hyperparameters $\lambda_M$ and $\lambda_T$. We empirically choose different combinations of $\lambda_M$ and $\lambda_T$, with $\lambda_M$ ranging from 1 to 10 and $\lambda_T$ from 1 to 5. We train several experiments from scratch with the same training settings introduced in Section 4.1. We evaluate each model during, and the variation curves of FID, R-precision, and Skating ratio are shown in Figure 4, where the mean of each metric after training becomes stable is reported top right in the legend. The quantitative results of FID, R-precision, and Skating ratio at the point where FID is the lowest are reported in Table 5. It is observed that all the combinations outperform the baseline MDM by a large margin, further suggesting the importance of allowing the text encoder to learn optimal text embeddings autonomously. As $\lambda_M$ increases from 1 to 10, the overall FIDs and Skating ratio become better. Increasing the value of $\lambda_T$ can lead to better text-motion matching precision, while the FID and Skating ratio become worse. In summary, $\lambda_M$ determines the motion quality and realism, and $\lambda_T$ determines the textual alignment of motion. Since we consider motion quality to be the most important (FIDs and Skating ratio), we choose $\lambda_M = 10$ and $\lambda_T = 2$ as our default setting ( the red line in Figure 4).

## 4.5 DATA EFFICIENCY

Our AdaQF can achieve competitive results only when trained with a small proportion of the training set. We validate the data efficiency by comparing AdaQF with three representative methods (MDM (Tevet et al., 2023), MoMask (Guo et al., 2024), SALAD (Hong et al., 2025), and our approach ) trained with 500, 1000, 2000, 4000, 6000, and 8000 training samples. The primary metrics, FID and R-Precision, are plotted as curves in Figure 5. MDM achieves competitive FIDs with 1000 and 2000 training samples, but its overall R-Precision stays relatively low at around 70%. MoMask exhibits

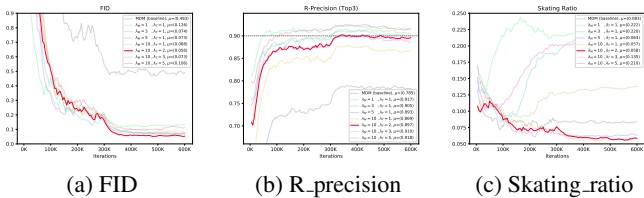

|                | (a) FID | (b) R_precision | (c) Skating_ratio |
|---|---|---|---|

Figure 4: Ablation studies of the hyperparameter $\lambda_M$ and $\lambda_T$. The curves of FID, R-Precision, and Skating Ratio during training. $\mu$ denotes the mean value after 400K iterations. The red line denotes our default setting.

| Method | FID ↓ | R-precision ↑ (Top-3) | Skating Ratio ↓ |
|---|---|---|---|
| MDM (baseline) | 0.357 | 0.712 | 0.096 |
| $\lambda_M = 1$ , $\lambda_T = 1$ | 0.099 | 0.917 | 0.225 |
| $\lambda_M = 3$ , $\lambda_T = 1$ | 0.047 | 0.905 | 0.227 |
| $\lambda_M = 5$ , $\lambda_T = 1$ | 0.048 | 0.893 | 0.064 |
| $\lambda_M = 10, \lambda_T = 1$ | 0.052 | 0.867 | **0.055** |
| $\lambda_M = 10, \lambda_T = 2$ | **0.039** | 0.900 | 0.062 |
| $\lambda_M = 10, \lambda_T = 3$ | 0.054 | 0.905 | 0.135 |
| $\lambda_M = 10, \lambda_T = 5$ | 0.081 | **0.927** | 0.208 |

Table 5: Ablation studies of the hyperparameter $\lambda_M$ and $\lambda_T$. Each row reports the results at the point where the FID is lowest.

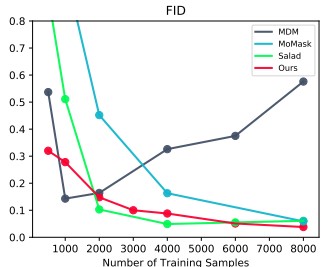

(a) The best FID curves under different training sample sizes.

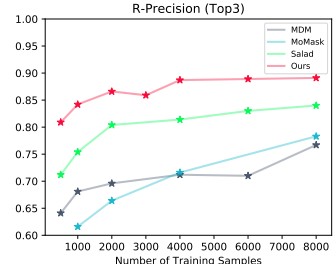

(b) The R-Precision curves under different training sample sizes.

Figure 5: The FID and R-Precision curves of MDM (Tevet et al., 2023), MoMask (Guo et al., 2024), SALAD (Hong et al., 2025), and our method, trained with different numbers of training samples. The size of the training set of HumanML3D is 23384.

steady improvements in both metrics as the training size increases from 500 to 8000; however, its performance remains weak below 2000 samples, suggesting that the VQ-VAE-based architecture requires a sufficient number of examples to learn the mapping from text to motion tokens. SALAD consistently maintains the second-highest R-Precision. Its FID is unsatisfactory below 2000 training samples but improves significantly beyond that threshold, which can be attributed to the latent-diffusion paradigm and its skeleton-aware attention. In contrast, our method achieves competitive FIDs and the highest R-Precision across all training sizes. Even with only 500 training samples, it attains the best FID and R-Precision simultaneously, highlighting its strong data efficiency. We provide more experiments with less training samples in APPENDIX A.1.

## 5 CONCLUSION

In this work, we introduced AdaQF, a feedback-driven framework that autonomously and efficiently adapts CLIP to text-to-motion generation. By providing motion quality feedback to AdaQF using motion and text consistency constraints, AdaQF enables the text encoder to learn optimal text embeddings and achieve a mutually adaptive state with the diffusion model. This co-optimization produces AdaCLIP, a motion-aware text encoder that is injected with motion-specific priors. Comprehensive experiments on HumanML3D and KIT-ML demonstrate that AdaQF achieves state-of-the-art results in terms of FID and R-Precision, converges substantially faster than existing baselines, and remains robust even with limited training data. We further demonstrate that AdaCLIP can be transferred into diverse generative paradigms (e.g., VQ-VAE-based and latent-diffusion-based) without changing other training settings, consistently improving generation quality and training efficiency.

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

# A APPENDIX

## A.1 THE PERFORMANCE WITH DIFFERENT NUMBERS OF TRAINING SAMPLES

In this section, we evaluate AdaQF and MDM under a few-shot learning setting. For AdaQF, we first train the text and motion extractors on the full training set (including 23384 samples). The extractor's training takes about 10 minutes on a single GPU. Then we train AdaQF with limited amounts of data. We still use the entire test set for evaluation, including 4383 samples. For the training scheduler, we train the networks for a maximum of 200 K iterations. The initial learning rate is 1e-4 and degrades by 0.1 at 100K iterations.

**Performances of AdaQF with 500, 1000, and 2000 training samples.** We first evaluate AdaQF on very small training sets of 500, 1000, and 2000 samples. Figure 6 shows the training curves of FID and R-Precision, illustrating how these metrics evolve during training, and Table 6 reports the quantitative results at the point where FID reaches its minimum. In the early stage of training, using fewer samples often leads to faster convergence. However, with only 500 training samples, clear signs of overfitting emerge. As the training set increases, the overfitting gradually diminishes and disappears at around 2000 training samples (roughly 10% of the full training set). When trained on 2000 samples, AdaQF converges markedly faster than the baseline MDM and achieves both lower FIDs and higher R-Precision.

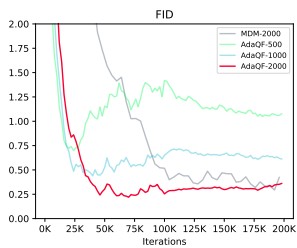

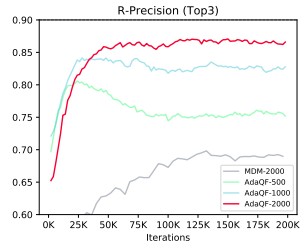

| Method | FID ↓ | R-precision ↑ (Top-3) | Skating Ratio ↓ |
|---|---|---|---|
| MDM-2000 | 0.176 | 0.696 | **0.104** |
| AdaQF-500 | 0.320 | 0.809 | 0.179 |
| AdaQF-1000 | 0.278 | 0.842 | 0.140 |
| AdaQF-2000 | **0.148** | **0.866** | 0.111 |

(a) FID during training.  (b) R-Precision during training.

Table 6: Quantitative results on small sample training.

Figure 6: Experiments of different training samples under a few-sample training setting. The curves include FID, R-Precision, and Skating Ratio during training.

**Performances comparison under 10% training samples.** Here we plot the FIDs and R-Precision when using only 2000 training samples (approximately 10%) in Figure 7. We observe that MoMask exhibits severe overfitting as training proceeds, whereas the three diffusion-based methods (MDM, SALAD, and our AdaQF) remain stable. We attribute this to the discrete tokenization used in VQ-VAE models, which limits representation flexibility and magnifies the impact of data scarcity. When samples are few, the pretrained codebook is insufficiently covered and the model quickly memorizes the training data. By contrast, diffusion-based methods model motions in a continuous space with noise perturbation and iterative denoising, which naturally act as a form of regularization and make them more robust under limited data.

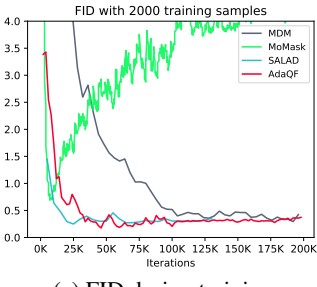

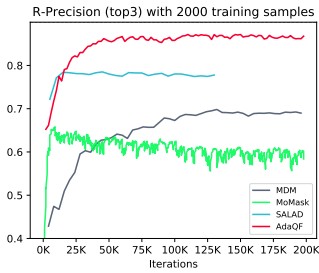

(a) FID during training.  (b) R-Precision during training.

Figure 7: The FID and R-Precision curves of MDM (Tevet et al., 2023), MoMask (Guo et al., 2024), SALAD (Hong et al., 2025), and our method, trained with 2000 training samples (approximately 10%). The size of the training set of HumanML3D is 23384.

## A.2 ADACLIP FROM HUMANML3D CAN BE APPLIED TO TRAIN MODELS ON KIT-ML.

Here we present the first attempt to transfer the learned knowledge to a new dataset. Having first trained AdaCLIP on the HumanML3D dataset, we substitute it for the vanilla CLIP encoder and keep it frozen when training several models on KIT-ML. We conduct experiments with three representative motion generation models: MDM, MoMask, and SALAD. For each model, we evaluate three settings: (i) the vanilla model with the original CLIP encoder (ii) the model with AdaCLIP trained on HumanML3D (denoted by "‡"), and (iii) the model with AdaCLIP trained on KIT-ML (denoted by "§"). Figure 8 reports the FIDs under these settings. Models equipped with Ada-CLIP‡ consistently achieve markedly better FIDs than the vanilla counterparts, demonstrating that the motion–text knowledge learned from HumanML3D transfers effectively to KIT-ML. By contrast, AdaCLIP§ yields slightly lower FIDs than AdaCLIP‡, which is expected given that KIT-ML is much smaller than HumanML3D, leading to weaker motion priors learned during training.

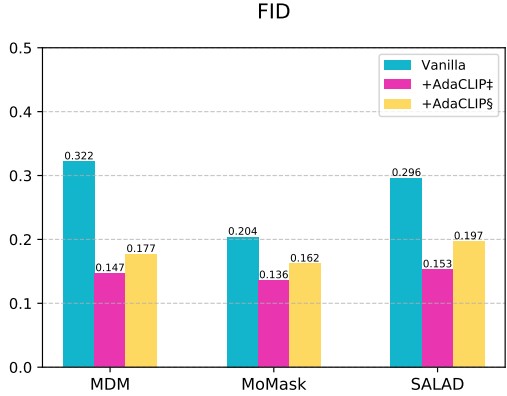

Figure 8: The FIDs of the vanilla model, plus the AdaCLIP finetuned on HumanML3D dataset ( with symbol "‡"), and the AdaCLIP finetuned on KIT-ML dataset ( with symbol "§").

## A.3 DISCUSSION ABOUT OUR PROPOSED EXTRACTORS AND OFFICIAL EVALUATORS.

T2M (Guo et al., 2022) trains a pair of text extractor and motion extractor with a conventional contrastive learning objective (Hadsell et al., 2006) as evaluators, where positive text–motion pairs are pulled closer and negative pairs are pushed apart. However, we argue that this strategy suffers from two limitations for constructing a unified embedding space. First, negative pairs are typically sampled randomly within a batch, which inevitably leads the model not to attend to the informative negative pairs simultaneously. Second, the reliance on a distance-based loss implicitly assumes that cross-modal embeddings should align in terms of absolute distance. However, text and motion embeddings are inherently heterogeneous modalities. The norm of the embeddings is modality-specific and contains the specific characteristics of each modality. Thus, applying a distance constraint may cause the text and motion embeddings to deviate from their respective semantic spaces. A qualified extractor for providing motion quality feedback should focus on capturing semantic similarity rather than absolute distance alignment. We report the quantitative results on using our trained extractors and those from (Guo et al., 2022). Result using the evaluators from Guo et al. (2022) achieves slightly worse R-Precision, but FID is lower than ours by a large margin.

| Methods | R Precision↑ | | | FID↓ | MultiModal Dist↓ |
| --- | --- | --- | --- | --- | --- |
| | Top 1 | Top 2 | Top 3 | | |
| with our extractors | $0.627^{\pm.004}$ | $0.811^{\pm.003}$ | $0.888^{\pm.002}$ | $0.039^{\pm.001}$ | $2.382^{\pm.010}$ |
| with evaluators from (Guo et al., 2022) | $0.611^{\pm.003}$ | $0.803^{\pm.002}$ | $0.871^{\pm.001}$ | $0.103^{\pm.003}$ | $2.407^{\pm.011}$ |

Table 7: Quantitative results on using our trained extractors and those from (Guo et al., 2022) on the HumanML3D testing set.

## A.4 THE RETRIEVAL PERFORMANCE OF OUR PROPOSED TEXT EXTRACTOR AND MOTION EXTRACTOR.

To illustrate the effectiveness of our text and motion extractors, we evaluate them on the text-to-motion and motion-to-text retrieval tasks. We adopted the commonly used metrics Recall (denoted as R@k for brevity) to compare, which include R@1,2,3,5,10. Higher Recall values indicate better performance. The retrievals are conducted on the test sets of the HumanML3D and KIT-ML benchmarks. Our results for both text-motion and motion-text retrieval are summarized in Table 8.

| Methods | Text-Motion Retrieval (%) ↑ | | | | | Motion-Text Retrieval (%) ↑ | | | | |
|---|---|---|---|---|---|---|---|---|---|---|
| | R@1 ↑ | R@2 ↑ | R@3 ↑ | R@5 ↑ | R@10 ↑ | R@1 ↑ | R@2 ↑ | R@3 ↑ | R@5 ↑ | R@10 ↑ |
| **HumanML3D** | | | | | | | | | | |
| TEMOS (Petrovich et al., 2022) | 42.40 | 53.52 | 61.14 | 70.96 | 84.15 | 39.96 | 53.49 | 61.79 | 72.40 | 85.89 |
| T2M (Guo et al., 2022) | 52.48 | 71.05 | 80.65 | 89.66 | 96.58 | 52.00 | 71.21 | 81.11 | 89.87 | 96.78 |
| TMR (Petrovich et al., 2023) | 67.16 | 81.32 | 86.81 | 91.43 | 95.36 | 67.97 | 81.20 | 86.35 | 91.70 | 95.27 |
| LAMP (Li et al., 2025) | $67.18^{\pm.5}$ | $81.9^{\pm0.4}$ | $87.04^{\pm0.3}$ | $92.0^{\pm0.2}$ | $95.73^{\pm0.2}$ | $68.02^{\pm0.3}$ | $82.1^{\pm0.3}$ | $87.5^{\pm0.3}$ | $92.2^{\pm0.3}$ | $96.9^{\pm0.3}$ |
| Ours | $\mathbf{69.15}^{\pm.004}$ | $\mathbf{84.43}^{\pm.002}$ | $\mathbf{90.21}^{\pm.002}$ | $\mathbf{94.89}^{\pm.002}$ | $\mathbf{98.35}^{\pm.001}$ | $\mathbf{69.0}^{\pm.003}$ | $\mathbf{84.77}^{\pm.002}$ | $\mathbf{90.52}^{\pm.003}$ | $\mathbf{95.26}^{\pm.002}$ | $\mathbf{98.74}^{\pm.001}$ |
| **KIT-ML** | | | | | | | | | | |
| T2MOS (Petrovich et al., 2022) | 43.88 | 58.25 | 67.00 | 74.00 | 84.75 | 41.88 | 55.88 | 65.62 | 75.25 | 85.75 |
| T2M (Guo et al., 2022) | 42.25 | 62.62 | 75.12 | 87.50 | 96.12 | 39.75 | 62.75 | 73.62 | 86.88 | 95.88 |
| TMR (Petrovich et al., 2023) | 49.25 | 69.75 | 78.25 | 87.88 | 95.00 | 50.12 | 67.12 | 76.88 | 88.88 | 94.75 |
| LAMP (Li et al., 2025) | $\mathbf{52.5}^{\pm0.7}$ | $\mathbf{74.8}^{\pm0.5}$ | $\mathbf{84.7}^{\pm0.5}$ | $92.7^{\pm0.3}$ | $97.6^{\pm0.3}$ | $\mathbf{54.0}^{\pm0.005}$ | $\mathbf{75.3}^{\pm0.5}$ | $84.4^{\pm0.4}$ | $92.2^{\pm0.2}$ | $97.6^{\pm0.2}$ |
| Ours | $51.33^{\pm.006}$ | $73.13^{\pm.004}$ | $83.62^{\pm.006}$ | $\mathbf{92.96}^{\pm.006}$ | $\mathbf{98.04}^{\pm.002}$ | $52.54^{\pm.010}$ | $74.16^{\pm.005}$ | $\mathbf{84.73}^{\pm.006}$ | $\mathbf{93.57}^{\pm.004}$ | $\mathbf{98.68}^{\pm.002}$ |

Table 8: Text-motion (**left**) and motion-text (**right**) retrieval benchmark on the HumanML3D and KIT-ML.

## A.5 THE CURVES OF DIFFERENT METRICS OF ABLATIONS ON PROPOSED COMPONENTS.

To more clearly illustrate the trends of FID, R-Precision, Skating Ratio, and MSE loss throughout the training process, we present learning curves depicting their evolution during training in Figure 9. All curves are smoothed using a sliding average with a coefficient of 0.8 for better visualization. The advantages of our method are not only reflected in the generation metrics but also in the physics-based Skaring Ratio. Using only MSE Loss leads the network to "mechanically" memorize the locations and rotations of each joint at each frame, instead of the high-level semantics. Even though our method keeps the bigger MSE loss, it achieves significant performance

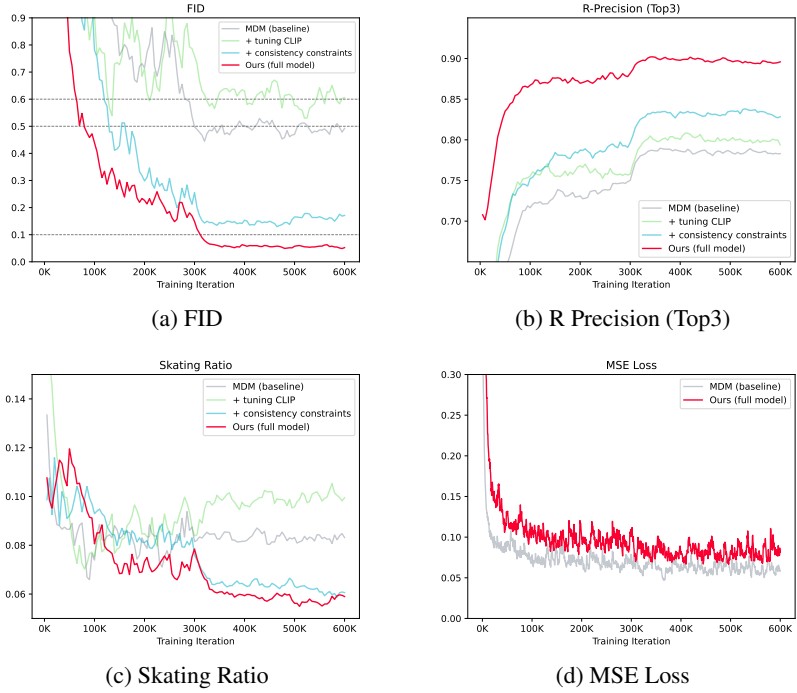

(a) FID

(b) R Precision (Top3)

(c) Skating Ratio

(d) MSE Loss

Figure 9: The curves of the FID, R-Precision, Skating Ratio, and MSE Loss during training of ablations on proposed components.

A.6 DISCUSSIONS ON THE MOST RELATED WORKS.

The most related to our AdaQF is MoCLIP (Maldonado et al., 2025) and LAMP (Li et al., 2025), both of which are dedicated to a better text encoder for text-to-motion generation. Here, we discuss our AdaQF with them from two aspects: motivation and technique.

**Regarding motivation**, MoCLIP and LAMP share the same motivation that the CLIP text encoder is limited in text-to-motion due to its pretraining on static image-text pairs. Both design a text encoder better suited for text-to-motion. Although our work shares the general goal of enhancing motion-aware semantics in text embeddings, our perspective is fundamentally different. We argue that the effectiveness of a text encoder for motion generation should not only be judged by representation-level alignment objectives it is trained on, but by the quality of the output motion.

**Technical innovation.** MoCLIP trains a new CLIP text encoder through contrastive learning and distillation. LAMP trains the LAMP text encoder via four pretrain tasks (motion-text contrastive learning, motion-text matching, motion-grounded text generation, and text-grounded motion generation) and requires several auxiliary classifier heads for training. Both methods plug their pretrained text encoders into existing motion generators (e.g., MoMask (Guo et al., 2024)) to evaluate the performance on text-to-motion generation. In contrast, our AdaQF jointly optimizes the text encoder (AdaCLIP) and the motion diffusion model under motion quality feedback, in which the text embeddings are continuously adjusted based on the quality of the output motions. This is the key technical innovation that distinguishes our method from previous works. To compare the transferability of our AdaCLIP and the two most related works, we provide the quantitative comparison results in Table 9, where all of them use MoMask as the backbone and substitute the original CLIP with their respective trained text encoder. The results of MoCLIP and LAMP are adopted directly from their paper. For LAMP, we use the quantitative metrics tested with bidirectional masked generation. We do not consider the Decoder-only strategy proposed by LAMP, as it differs from MoMask's setting. We also provide the results of various levels of residual quantization (RQ) in Table 9. Even with just one RQ level, our method surpasses MoMask using all RQ levels.

**Advantages.** We elaborate on the advantages of our method compared to MoCLIP (Maldonado et al., 2025) and LAMP (Li et al., 2025).

- Our AdaQF achieves state-of-the-art performance among the diffusion-based methods. It achieves approximately 4x faster convergence than baseline, and achieves >90% improvement on FID and does not introduce additional parameters. Compared to LAMP, our methods provide AdaQF (a full text-to-motion generation framework) and AdaCLIP (a plug-and-play module with high transferability). However, LAMP focuses on text-motion pretraining and does not constitute a full text-to-motion generation pipeline.

- Regarding transferability into the same MoMask network, using our AdaCLIP achieves better performance than using LAMP or MoCLIP in Table 9. For LAMP, we use the quantitative metrics tested with bidirectional masked generation. We do not consider the Decoder-only strategy proposed by LAMP, as it differs from MoMask's setting. In addition to the quantitative advantages, MoMask+AdaCLIP achieves approximately 8x faster training speed than vanilla MoMask. Transferability is also validated on another two types of generative frameworks: diffusion-based, latent-diffusion-based, and VQVAE-based.

- Regarding the cross-dataset transferability from HumanML3D to KIT-ML using MoMask as baseline network, our AdaCLIP achieves better performance than LAMP, while LAMP exhibits slightly worse performance than the baseline.

- Our extractors exhibit better retrieval precision (as in Table 8) with a significantly simpler training pipeline compared to LAMP. For LAMP training, four pretraining tasks are required, and auxiliary classifier heads are required for motion-text matching, motion-grounded text generation, and text-grounded motion generation tasks. In contrast, our extractors are trained with two simple loss functions, which reduces the complexity of multi-task training. On the HumanML3D dataset (23,384 training samples), our extractor's training takes about 10 minutes on a single GPU. Although LAMP reports strong results, its pretraining computational cost (in terms of time and memory) is not disclosed in the original paper, and its code has not been released.

| Methods | R Precision↑ | | | FID↓ | MultiModal Dist↓ |
|---|---|---|---|---|---|
| | Top 1 | Top 2 | Top 3 | | |
| MoMask (Guo et al., 2024) | $0.521^{\pm.002}$ | $0.713^{\pm.002}$ | $0.807^{\pm.002}$ | $0.045^{\pm.002}$ | $2.958^{\pm.008}$ |
| MoMask+MoCLIP (Maldonado et al., 2025) | $0.533^{\pm.003}$ | $0.730^{\pm.002}$ | $0.823^{\pm.001}$ | $0.047^{\pm.002}$ | $2.868^{\pm.006}$ |
| MoMask+LAMP (Li et al., 2025) | $0.550^{\pm.003}$ | $0.741^{\pm.002}$ | $0.834^{\pm.001}$ | $0.042^{\pm.002}$ | $2.978^{\pm.006}$ |
| MoMask+AdaCLIP-R5 | $\mathbf{0.554}^{\pm.003}$ | $\mathbf{0.751}^{\pm.002}$ | $\mathbf{0.842}^{\pm.003}$ | $\mathbf{0.040}^{\pm.002}$ | $\mathbf{2.771}^{\pm.009}$ |
| ├──R3 | $0.546^{\pm.003}$ | $0.740^{\pm.002}$ | $0.831^{\pm.003}$ | $0.042^{\pm.002}$ | $2.825^{\pm.009}$ |
| ├──R1 | $0.537^{\pm.004}$ | $0.733^{\pm.003}$ | $0.827^{\pm.003}$ | $0.041^{\pm.002}$ | $2.845^{\pm.009}$ |
| ├──R0 | $0.526^{\pm.009}$ | $0.721^{\pm.001}$ | $0.815^{\pm.002}$ | $0.047^{\pm.005}$ | $2.916^{\pm.006}$ |

Table 9: Quantitative results on using our trained extractors and those from (Guo et al., 2022) on the HumanML3D testing set.

### A.7 VISUAL COMPARISONS OF WHETHER USING LORA-BASED FINETUNING.

We provide two visual comparisons of using LoRA-based finetuning or full-parameter finetuning in Figure 10, demonstrating that LoRA-based finetuning maintains most of the original CLIP's general text understanding ability.

*A person is **holding something** in front of him and **swings** to the left.*

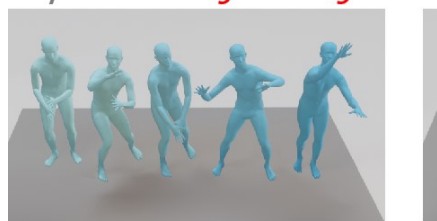 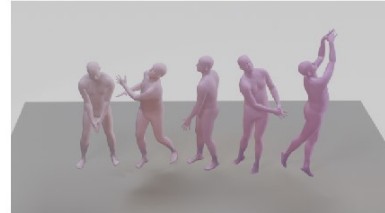

(a) Sample 1

*A person walks in a clockwise direction **to his right**.*

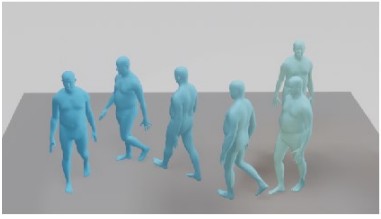 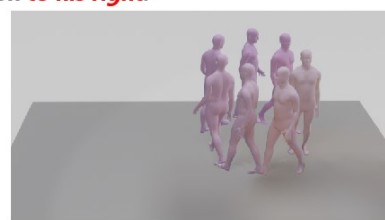

(b) Sample 2

Figure 10: Visual comparisons on using LoRA-based finetuning or full-parameter finetuning. Left figures correspond to LoRA-based finetuning; right figures correspond to full-parameter finetuning.

### A.8 USER STUDY

We further conduct a user study to compare the realism and semantic alignment of our method with previous methods: MDM (Tevet et al., 2023), MoMask (Guo et al., 2024), and SALAD (Hong et al., 2025). We randomly sample 20 texts for sampling. 25 participants were asked to rank videos generated by four methods based on their realism and semantic alignment. Results in Figure 11 show that our method clearly outperforms others in both realism and semantic alignment.

### A.9 MORE VISUAL SAMPLES OF OUR METHOD.

We provide several additional visual samples of text-to-motion generation in Figure 12.

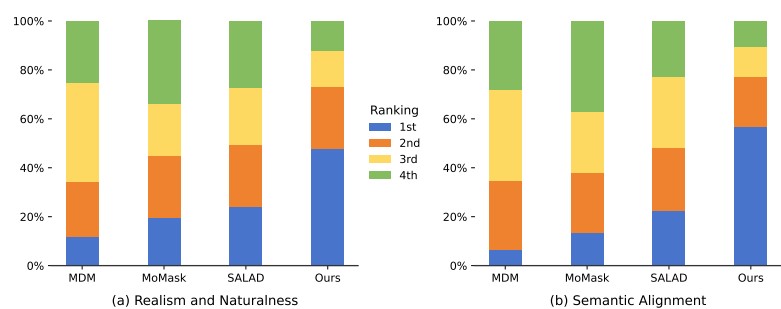

Figure 11: User study comparing our method with MDM, MoMask, and SALAD.

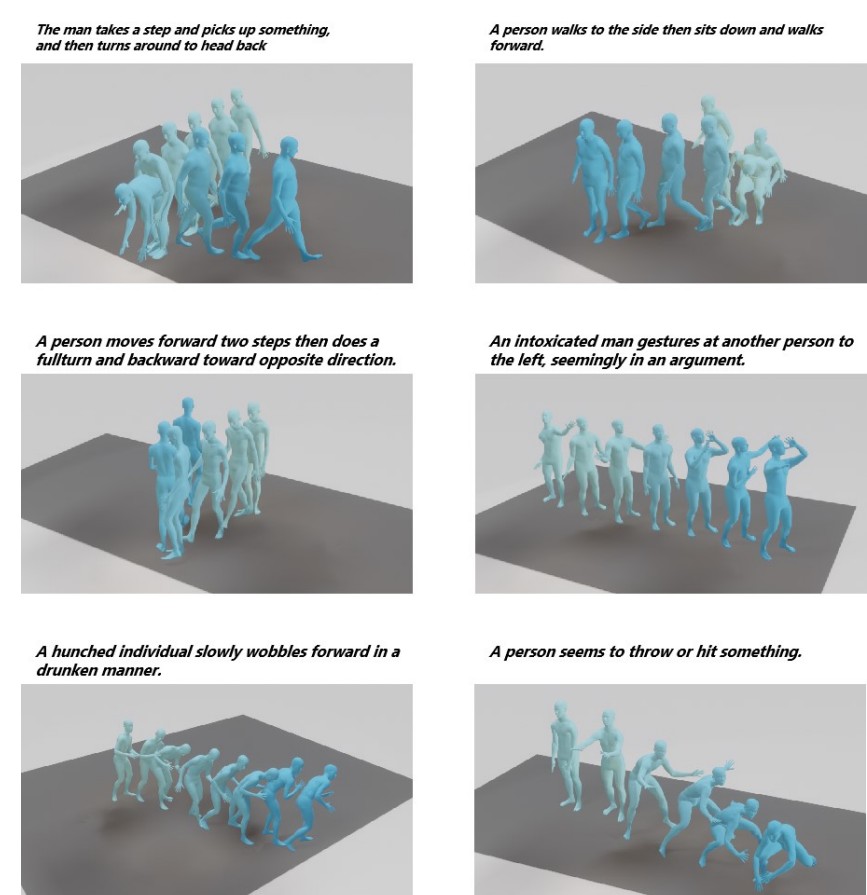

Figure 12: Visual samples of text-to-motion generation on HumanML3D testsing set.

## A.10 USAGE OF LARGE LANGUAGE MODELS.

We used large language models (LLMs) solely as a general-purpose assistive tool for assisting paper writing (e.g., improving grammar, clarity, and style). The research ideas, experimental design, implementation, analyses, and all scientific claims in this paper are entirely from the author's discussion. The LLM did not contribute to the conception of the research, the development of methods, or the interpretation of results. We take full responsibility for the content of this paper.

