# OpenReview forum: "High-Fidelity Human Motion Generation with Motion Quality Feedbacks"
_ICLR.cc/2026/Conference — Submitted to ICLR 2026_

### Official Review · Reviewer_3KCs · 2025-10-29

**Soundness:** 3
**Presentation:** 3
**Contribution:** 2
**Rating:** 4
**Confidence:** 4

**Summary:**

The paper presents AdaQF to improve the performance of text2motion generation. One common issue with CLIP encoder in the T2M task is: CLIP is trained for static visual semantics and is bad at temporal information. This paper proposes to fine-tune CLIP text encoder with LoRA rather than freeze it when training the text2motion diffusion models. It introduces two additional loss terms in the semantics space via pretrained text and motion contrastive encoders. Experiments show significantly improved performance on common benchmarks.

**Strengths:**

1. The proposed method is easy to follow and makes a lot of sense to me. The performance improvement is significant. A simple yet effective approach should always be preferable.
2. The proposed method also shows faster convergence and transferability, which might potentially have a larger impact on the broader research community by using similar techniques or pretrained text encoder.
3. The qualitative results look very good compared to previous SOTA methods.

**Weaknesses:**

1. For the terminology, I cannot agree that the proposed method should be described as "feedback". For "feedback", I would rather expect an iterative loop that improves the model from the old version. The proposed method adds additional loss terms to the common supervision. Though it's simple, the experiment shows its effectiveness. I don't think it's proper to wrap it with a fancy term that people think of RLHF, which has nothing to do with the proposed method. I neither agree that it's about "quality" rather than semantics.
2. The paper should discuss more related work that uses additional loss terms computed in latent semantics space, such as LaMP[1]. The experiment should also include more SOTA methods or explain the reason why not reported those methods.

[1] Li, Zhe, et al. "LaMP: Language-Motion Pretraining for Motion Generation, Retrieval, and Captioning." The Thirteenth International Conference on Learning Representations.

**Questions:**

1. I wonder how the proposed method is different from previous work, i.e. LaMP [1], which also trains the text encoder along with the generator, as well as semantic latent space supervision via some contrastive encoder. What's the advantage of proposed method compared to the closely related work?
2. What's the necessity of LoRA finetuning instead of full parameter fine-tuning? Will full parameter fine-tuning lead to even better performance?
3. From Table 4, it seems that tuning the CLIP without the proposed two loss terms will degrade the performance. I wonder what's the possible reason.
4. If the additional two loss term is the actual effective design, I'm curious how the vanilla MDM could benefit from the proposed loss terms. So, I would expect to see the result of vanilla MDM trained with the proposed "consistency constraints".

[1] Li, Zhe, et al. "LaMP: Language-Motion Pretraining for Motion Generation, Retrieval, and Captioning." The Thirteenth International Conference on Learning Representations.

---

> ### Author Response · Authors · 2025-11-21
> **Response to Reviewer 3KCs (1/2)**
>
> Thank you for your valuable feedback. Below, we address your questions and comments.
>
>
> ```
> Q1: The explanation of the notion "motion quality feedback".
> ```
>
> We appreciate the reviewer's concern regarding the use of the term "feedback". We want to clarify that the terminology is not inspired by RLHF or iterative policy improvement. Instead, **our usage follows the classical notion of feedback in automatic control systems**, where the system's output is mapped back to the reference signal to compute an error that guides parameter adjustment.
> In our AdaQF, AdaCLIP serves as the "controller": it receives the text prompt as the "reference signal" and produces the text embedding that conditions the generation process. The motion diffusion model serves as the "actuator", which makes motion sequences based on this conditioning signal. The proposed consistency constraints then act as the "feedback": they compare the semantic representation of the generated motion with the text embedding and compute a feedback error. This feedback signal is backpropagated to update the parameters of both the "controller" (AdaCLIP) and the "actuator" (diffusion model), thus forming a semantic feedback loop during training.
>
> ```
> Q2: The necessity of LoRA finetuning. Will full parameter fine-tuning lead to better performance?
> ```
>
> We chose to use LORA for fine-tuning based on the following two reasons. First, we hope to preserve the pretrained prior of the original CLIP text encoder while adapting to the text-to-motion task. Second, the HumanML3D and KIT-ML datasets are much smaller than those used for training CLIP, and using LORA can significantly reduce the risk of overfitting.
> To validate this choice, we additionally conducted an experiment in which the entire CLIP text encoder and the motion diffusion model were jointly finetuned under our proposed consistency constraints. Full-parameter finetuning performs evidently worse than our LoRA-based tuning (FID: full, R@3: full tuning: 0.073, LoRA tuning: 0.039). We have reported the results below and added them in Ablation 5 of Table 4.
> This demonstrates that unrestricted updates tend to destabilize CLIP’s pretrained prior and cause overfitting. In contrast, LoRA provides a more stable and effective way to specialize the text encoder for motion generation.
>
> | Method | FID $\downarrow$ | R@3 $\uparrow$
> | :--- | :---: | :---: |
> | full tuning | $0.073^{±.002}$ | $0.873^{±.002}$ |
> | LoRA tuning | $0.039^{±.001}$ | $0.888^{±.002}$ |
>
>
>
>
> ```
> Q3: Explanation of why tuning the CLIP without the proposed two consistency constraints degrades the performance.
> ```
>
> We attribute this degradation to shortcut learning. When the CLIP text encoder is unfrozen but trained solely under the low-level MSE loss, the system of CLIP and diffusion minimizes reconstruction error rather than establishing a faithful text-to-motion mapping. As a consequence, the CLIP encoder tends to produce text embeddings that lose semantics, while the diffusion model simultaneously adapts to interpret these distorted embeddings. This leads to text embeddings that overfit to the training distribution and fail to generalize to unseen descriptions, ultimately harming generation quality.
> We have added this discussion to Section 4.4 in the revised version.
>
> ```
> Q4: The vanilla MDM trained with the consistency constraints.
> ```
>
> We want to clarify that this experiment was already presented in Table 4 of the initial submission. The term "w/o tuning CLIP" in the initial manuscript refers to training MDM with the two proposed consistency constraints. To clarify the performance of each component when applied to the baseline MDM, we reformatted Table 4 so that the first row represents the baseline MDM. Ablation 2 in Table 4 of the revised version corresponds to training vanilla MDM with consistency constraints.
>
> ```
> Q5: Include more state-of-the-art methods in experimental comparisons.
> ```
> We have added the quantitative comparisons with LAMP [1], HGM^3 [2], and DisCoRD [3] on HumanML3D and KIT-ML benchmarks.
>
> [1] LAMP: LANGUAGE-MOTION PRETRAINING FOR MOTION GENERATION, RETRIEVAL, AND CAPTIONING. In ICLR 2025
>
> [2] HGM^3: HIERARCHICAL GENERATIVE MASKED MOTION MODELING WITH HARD TOKEN MINING. In ICLR 2025
>
> [3] DisCoRD: Discrete Tokens to Continuous Motion via Rectified Flow Decoding. In ICCV 2025

---

> ### Author Response · Authors · 2025-11-21
> **Response to Reviewer 3KCs (2/2)**
>
> ```
> Q6: The difference and advantage compared with LAMP
> ```
> We first clarify that LAMP **does not "train the text encoder along with the generator"**. LAMP first trained a text encoder and then applied it to existing text-to-motion methods.
> We discuss the difference compared with LAMP below. We have added comparisons in Tables 1 and 2 and discussions in APPENDIX A.6.
>
> **Regarding motivation,** LAMP shares the motivation that the CLIP text encoder is limited in text-to-motion due to its pretraining on static image-text pairs. LAMP designs a text encoder better suited for text-to-motion. Although our work shares the general goal of enhancing motion-aware semantics in text embeddings, our perspective is fundamentally different. We argue that the effectiveness of a text encoder for motion generation should not only be judged by representation-level alignment objectives it is trained on, but by the quality of the output motion.
>
> **Technical innovation.** LAMP trains the LAMP text encoder via four pretrain tasks (motion-text contrastive learning, motion-text matching, motion-grounded text generation, and text-grounded motion generation), which serve as a plug-and-play module in other text-to-motion methods. In contrast, our AdaQF jointly optimizes AdaCLIP and the motion diffusion model with motion-quality feedback, in which the text embeddings are continuously adjusted based on the quality of the generated motions. This is the key technical innovation that distinguishes our method from LAMP.
>
>
> **Advantages.**
>
> 1. Our AdaQF achieves state-of-the-art performance among the diffusion-based methods. It converges approximately 4x faster than baseline, and achieves $>$90\% improvement in FID and does not introduce additional parameters. Compared to LAMP, our methods include AdaQF (a full text-to-motion generation framework) and AdaCLIP (a plug-and-play module with high transferability). However, LAMP focuses on text-motion pretraining and does not constitute a full text-to-motion generation pipeline.
>
> 2. Regarding transferability into the same MoMask network, using AdaCLIP achieves better performance than using LAMP. In addition to the quantitative advantages, MoMask+AdaCLIP achieves approximately 8x faster training speed. We report the comparisons on HumanML3D below. For LAMP, we use the quantitative metrics tested with bidirectional masked generation. We do not consider the Decoder-only strategy proposed by LAMP, as it differs from MoMask’s setting. Our AdaCLIP also demonstrates transferability to MDM and SALAD with clear improvements, as shown in Table 3.
>
>
> | Method | R@Top1 $\uparrow$ | R@Top2 $\uparrow$ | R@Top3 $\uparrow$ | FID $\downarrow$ | MultiModal Dist $\downarrow$
> | :--- | :---: | ---: | :---: | :---: | :---:
> | MoMask | $0.521^{±.002}$ | $0.713^{±.002}$ | $0.807^{±.002}$ | $0.045^{±.002}$ | $2.958^{±.008}$ |
> | MoMask+LAMP  | $0.550^{±.002}$ | $0.741^{±.002}$ | $0.834^{±.002}$ | $0.042^{±.002}$ | $2.978^{±.006}$ |
> | MoMask+AdaCLIP | $0.554^{±.003}$ | $0.751^{±.002}$ | $0.842^{±.003}$ | $0.040^{±.002}$ | $2.771^{±.009}$ |
>
>
> 3. Regarding the cross-dataset transferability from HumanML3D to KIT-ML using MoMask as baseline network, AdaCLIP achieves better than LAMP, while LAMP exhibits slightly worse than the baseline. We report the comparisons on KIT-ML below. The used LAMP and AdaCLIP are trained on HumanML3D.
>
> | Method | R@Top1 $\uparrow$ | R@Top2 $\uparrow$ | R@Top3 $\uparrow$ | FID $\downarrow$ | MultiModal Dist $\downarrow$
> | :--- | :---: | :---: | :---: | :--- | :---:
> | MoMask | $0.433^{±.007}$ | $0.656^{±.005}$ | $0.781^{±.005}$ | $0.204^{±.011}$ | $2.779^{±.022}$ |
> | MoMask+LAMP $_{ML3D}$  | $0.423^{±.006}$ | $0.657^{±.005}$ | $0.771^{±.005}$ | $0.226^{±.012}$ (-10%) | $2.768^{±.022}$ |
> | MoMask+AdaCLIP $_{ML3D}$ | $0.437^{±.008}$ | $0.652^{±.010}$ | $0.785^{±.007}$ | $0.136^{±.006} $ (+66%) | $2.780^{±.025}$ |
>
> 4. Compared to LAMP, our extractors achieve higher retrieval precision and are more training-friendly than LAMP. We present the performance of text-to-motion retrieval below due to page limits, the whole comparison is included in APPENDIX Table 8. For LAMP training, four pretraining tasks and **several auxiliary classifier heads** are required. In contrast, our extractors are trained with two simple loss functions, which reduces the complexity of multi-task training.
> On the HumanML3D dataset (23384 training samples), our extractor's training **takes about 10 minutes** on a single GPU. Although LAMP reports strong results, its pretraining computational cost (in terms of time and memory) is not reported in the original paper.
>
> | Method | R@1 $\uparrow$ | R@2 $\uparrow$ | R@3 $\uparrow$ |  R@5 $\uparrow$ | R@10 $\uparrow$ |
> | :--- | :---: | :---: | :---: | :---: | :---:
> | T2M| 52.48 | 71.05 | 80.65 | 89.66| 96.58 |
> | LAMP | 67.18 | 81.90 | 87.04 | 92.00 | 95.73 |
> | Ours | 69.15 | 84.43 | 90.21 | 94.89 | 98.35 |

---

### Official Review · Reviewer_FR6P · 2025-10-30

**Soundness:** 2
**Presentation:** 2
**Contribution:** 1
**Rating:** 2
**Confidence:** 5

**Summary:**

This paper proposes AdaQF, which accomplishes the motion generation task by fine-tuning the pre-trained CLIP in the form of LoRA and introducing a semantic consistency regularization loss. Experiments demonstrate that this method effectively improves the quality of generated motions and enhances the semantic information of the motions.

**Strengths:**

1.This paper identifies the inadequacy of CLIP in motion generation capabilities and improves the generation performance by fine-tuning CLIP.

2.It proposes MoCLIP to replace the pre-trained CLIP, and introduces a semantic consistency loss by training a motion extractor and a text extractor, thereby enhancing both the generation quality and the semantic information of the generated motions.

3.The experiments are relatively comprehensive, and the structure of the paper is clear.

4.Some video demos are provided, making the result presentation more clear.

**Weaknesses:**

1.My primary concern is that the novelty of this contribution is very limited. The inadequacy of CLIP's representational capability in motion generation has already been addressed in similar works, which have also proposed corresponding solutions (e.g., LaMP). However, this paper does not discuss this work, nor does it include it in the baselines.

2.The authors pre-train a Motion Extractor and a Text Extractor, use these two modules to generate corresponding embeddings, and calculate the similarity between these embeddings as the loss. In my opinion, this is an approach that treats a metric as a loss. A pre-trained evaluator should achieve the same effect, and the similarity is also closely related to the R-Precision metric.

3.The related works included in the baselines are insufficient; comparisons with more existing works are required.

4.Some visual demo images can be further added in the main text.

**Questions:**

1.Discuss the main differences from LaMP, compare with its experimental results, and elaborate on the advantages over it.

2.Although CLIP is a foundation model, its parameter count is not large. Why is fine-tuning performed in the form of LoRA, and would full-parameter tuning yield better results?

3.The authors claim that this optimization method retains CLIP's text understanding capability. Can this be proven?

4.In the ablation study, why is the FID score of the "without both" setting better than that of the "without consistency alone" setting?

5.In the appendix, I find it strange why the results of AdaQF trained without using KIT-ML are better than those trained on KIT-ML. Does this indirectly indicate that the method is only effective for the R-Precision metric? Because I believe it is an approach that optimizes the loss by adopting the metric calculation method.

---

> ### Author Response · Authors · 2025-11-21
> **Response to Reviewer FR6P (1/3)**
>
> We thank the reviewer for your thoughtful comments.
> ```
> Q1: The difference and advantage compared with LAMP
> ```
> We discuss the motivation, technical innovation, and advantages compared with LAMP below. We have added comparisons in Tables 1 and 2 and discussions in APPENDIX A.6.
>
> **Regarding motivation,** LAMP shares the motivation that the CLIP text encoder is limited in text-to-motion due to its pretraining on static image-text pairs. LAMP designs a text encoder better suited for text-to-motion. Although our work shares the general goal of enhancing motion-aware semantics in text embeddings, our perspective is fundamentally different. We argue that the effectiveness of a text encoder for motion generation should not only be judged by representation-level alignment objectives it is trained on, but by the quality of the output motion.
>
> **Technical innovation.** LAMP trains the LAMP text encoder via four pretrain tasks (motion-text contrastive learning, motion-text matching, motion-grounded text generation, and text-grounded motion generation), which serve as a plug-and-play module in other text-to-motion methods. In contrast, our AdaQF jointly optimizes AdaCLIP and the motion diffusion model with motion-quality feedback, in which the text embeddings are continuously adjusted based on the quality of the generated motions. This is the key technical innovation that distinguishes our method from LAMP.
>
>
> **Advantages.**
>
> 1. Our AdaQF achieves state-of-the-art performance among the diffusion-based methods. It converges approximately 4x faster than baseline, and achieves $>$90\% improvement in FID and does not introduce additional parameters. Compared to LAMP, our methods include AdaQF (a full text-to-motion generation framework) and AdaCLIP (a plug-and-play module with high transferability). However, **LAMP focuses on text-motion pretraining and does not constitute a full text-to-motion generation pipeline.**
>
> 2. Regarding transferability into the same MoMask network, using AdaCLIP achieves better performance than using LAMP. In addition to the quantitative advantages, MoMask+AdaCLIP achieves approximately 8x faster training speed. We report the comparisons on HumanML3D below. For LAMP, we use the quantitative metrics tested with bidirectional masked generation. We do not consider the Decoder-only strategy proposed by LAMP, as it differs from MoMask’s setting. Our AdaCLIP also demonstrates transferability to MDM and SALAD with clear improvements, as shown in Table 3.
>
>
> | Method | R@Top1 $\uparrow$ | R@Top2 $\uparrow$ | R@Top3 $\uparrow$ | FID $\downarrow$ | MultiModal Dist $\downarrow$
> | :--- | :---: | ---: | :---: | :---: | :---:
> | MoMask | $0.521^{±.002}$ | $0.713^{±.002}$ | $0.807^{±.002}$ | $0.045^{±.002}$ | $2.958^{±.008}$ |
> | MoMask+LAMP  | $0.550^{±.002}$ | $0.741^{±.002}$ | $0.834^{±.002}$ | $0.042^{±.002}$ | $2.978^{±.006}$ |
> | MoMask+AdaCLIP | $0.554^{±.003}$ | $0.751^{±.002}$ | $0.842^{±.003}$ | $0.040^{±.002}$ | $2.771^{±.009}$ |
>
>
> 3. Regarding the cross-dataset transferability from HumanML3D to KIT-ML using MoMask as baseline network, AdaCLIP achieves better than LAMP, while LAMP exhibits slightly worse than the baseline. We report the comparisons on KIT-ML below. The used LAMP and AdaCLIP are trained on HumanML3D.
>
> | Method | R@Top1 $\uparrow$ | R@Top2 $\uparrow$ | R@Top3 $\uparrow$ | FID $\downarrow$ | MultiModal Dist $\downarrow$
> | :--- | :---: | ---: | :---: | :---: | :---:
> | MoMask | $0.433^{±.007}$ | $0.656^{±.005}$ | $0.781^{±.005}$ | $0.204^{±.011}$ | $2.779^{±.022}$ |
> | MoMask+LAMP $_{ML3D}$  | $0.423^{±.006}$ | $0.657^{±.005}$ | $0.771^{±.005}$ | $0.226^{±.012}$ | $2.768^{±.022}$ |
> | MoMask+AdaCLIP $_{ML3D}$ | $0.437^{±.008}$ | $0.652^{±.010}$ | $0.785^{±.007}$ | $0.136^{±.006}$ | $2.780^{±.025}$ |
>
> 4. Compared to LAMP, our extractors achieve higher retrieval precision and are more training-friendly than LAMP. We present the text-to-motion retrieval performance below due to page limits, the whole comparison is included in APPENDIX Table 8. For LAMP training, four pretraining tasks and **several auxiliary classifier heads** are required. In contrast, our extractors are trained with two simple loss functions, which reduces the complexity of multi-task training.
> On the HumanML3D dataset (23384 training samples), our extractor's training **takes about 10 minutes** on a single GPU. Although LAMP reports strong results, its pretraining computational cost (time and memory) is not reported in the original paper.
>
> | Method | R@1 $\uparrow$ | R@2 $\uparrow$ | R@3 $\uparrow$ |  R@5 $\uparrow$ | R@10 $\uparrow$ |
> | :--- | :---: | :---: | :---: | :---: | :---:
> | T2M| 52.48 | 71.05 | 80.65 | 89.66| 96.58 |
> | LAMP | 67.18 | 81.90 | 87.04 | 92.00 | 95.73 |
> | Ours | 69.15 | 84.43 | 90.21 | 94.89 | 98.35 |

---

> ### Author Response · Authors · 2025-11-21
> **Response to Reviewer FR6P (2/3)**
>
> ```
> Q2:  Question regarding the use of embedding similarity as a training objective.
> ```
> The reviewer is concerned that the proposed loss may be related to the evaluation protocol, and wonders whether a pretrained evaluator would achieve the same effect.
> We clarify that using the pre-trained evaluator results in a clearly worse outcome. We propose consistency constraints based on the motivations we introduce below, which differs in the implementation from R-Precision. Our method does not "treat a metric as a loss".
>
> 1.**Using official evaluators performs worse.** If our method was designed to exploit the evaluation protocol, then training with the official evaluator as extractors should yield perfect results. Therefore, we conducted additional experiments using the official evaluator for training, which yield the opposite results (FID: ours default 0.039 vs official evaluator 0.103; see Appendix Table 7). This demonstrates that the performance improvements come from our extractors and consistency constraints rather than any metric-specific bias.
>
> 2.**Difference between our extractors and evaluation protocol.** The official evaluator is trained using Euclidean-distance-based loss functions, thus it can map all the motions into a compact embedding space. This is reflected by the FID score between two set of randomly truncated ground truth motion data (GT FID = 0.002), showing that it is well-suited for FID evaluation. While it is qualified for evaluating FID, the GT R-Precision Top3 is 79.7%, which might not sufficiently represent the alignment between texts and motions in ground truth data.
>
>
> In contrast, our extractors are trained with cosine-similarity-based InfoNCE losses. This training objective emphasizes directional similarity between text and motion embeddings without imposing magnitude constraints.  As a result, our extractors are not designed and not suitable for FID evaluation and naturally yield a much higher GT FID of 0.186 , but excel in text-to-motion retrieval (GT R@3 = 90.2%, Table 9). Therefore, our method doesn't treat the metric as a loss. Moreover, R-Precision is calculated using Euclidean distance, which provides a stricter constraint than cosine similarity.
>
> 3.**The motivation of Motion Consistency Constraint.** We believe that it is not optimal to judge the closeness between GT and predicted motion with only MSE loss during training. Even though several joints in motion slightly deviate from the GT at some frames, as long as the overall motion remains visually similar to GT, then this motion should not be assigned a significant loss value. We should provide a certain "degree of freedom" to the network to search for any similar motions to GT. Thus, we consider judging the similarity in a high-level embedding space.
>
> 4.**The motivation of our Text Consistency Constraint.** The text conditions are coarse-grained, and there is no strict one-to-one correspondence between text and GT motion. Many possible motions can match the text, rather than just the GT. Therefore, we consider allowing the network to learn the consistency within a high-level text and motion embedding space.
>
> 5.**Physics-based metric.** Instead of only the main metrics, we also include a metric called Skating Ratio in Table 4, which most text-to-motion methods do not provide. The Skating Ratio is a low-level metric that represents the proportion of frames in which foot skating occurs, which is computed independently of the evaluator
>
> 6.Figure 9 in the revised version confirms our analysis that using only MSE Loss causes the network to "mechanically" memorize the locations and rotations of each joint at every frame, rather than capturing high-level semantics. Even though our method has a higher MSE loss, it achieves significantly better generation performance.
>
> ```
> Q3: Add more comparisons with existing works.
> ```
> We have added the quantitative comparisons with LAMP [1], HGM^3 [2], and DisCoRD [3] on HumanML3D and KIT-ML benchmarks.
>
> [1] LAMP: LANGUAGE-MOTION PRETRAINING FOR MOTION GENERATION, RETRIEVAL, AND CAPTIONING. In ICLR 2025
>
> [2] HGM^3: HIERARCHICAL GENERATIVE MASKED MOTION MODELING WITH HARD TOKEN MINING. In ICLR 2025
>
> [3] DisCoRD: Discrete Tokens to Continuous Motion via Rectified Flow Decoding. In ICCV 2025
>
> ```
> Q4: Add more visual demo images.
> ```
>
> Thanks for this advice. We provided visual comparisons in Figure 3, and each of the four methods has four samples. We have added more visual images in Figure 12 in the APPENDIX of the revised version due to page limit.

---

> ### Author Response · Authors · 2025-11-21
> **Response to Reviewer FR6P (3/3)**
>
> ```
> Q5: Why is fine-tuning performed with LoRA, and would full-parameter tuning yield better results?
> ```
> We chose to use LORA for fine-tuning based on the following two reasons. First, we hope to preserve the pretrained prior of the original CLIP text encoder while adapting to the text-to-motion task. Second, the HumanML3D and KIT-ML datasets are much smaller than those used for training CLIP, and using LORA can significantly reduce the risk of overfitting.
> We conduct experiments on finetuning CLIP with full-parameter tuning. We report the results below, and have added this in Ablation 5 of Table 4. Full-parameter finetuning yields worse generation quality than our default LoRA-based tuning, by a clear margin.
>
> | Method | FID $\downarrow$ | R@3 $\uparrow$
> | :--- | :---: | :---: |
> | full tuning | $0.073^{±.002}$ | $0.873^{±.002}$ |
> | LoRA tuning | $0.039^{±.001}$ | $0.888^{±.002}$ |
>
>
> ```
> Q6: The authors claim that this optimization method retains CLIP's text understanding capability. Can this be proven?
> ```
>
> We believe that "this optimization method" you mentioned is our AdaQF. We primarily use LoRA to keep most pretrained parameters unchanged and maintain the pretrained prior, since these parameters are trained on enormous datasets. LoRA finetuning is empirically more stable and leads to better generation performance than full finetuning. To avoid ambiguity, we have revised the wording "retain CLIP’s general language understanding" into "maintain CLIP's pretrained prior" in the revised version.  要改
> We believe that "this optimization method" you mentioned is our AdaQF. We prove this claim via quantitative and qualitative comparisons. As answered in Q5, finetuning with LoRA performs clearly better than full-parameter finetuning. We also provide visual comparisons for using LoRA-based tuning or full-parameter tuning in Figure 10 in the revised submission.
>
> ```
> Q7: In the ablation study, why is the FID score of the "without both" setting better than that of the "without consistency alone" setting?
> ```
>
> We attribute this degradation to shortcut learning. When the CLIP text encoder is unfrozen but trained solely on the low-level MSE loss, the CLIP-diffusion system minimizes reconstruction error rather than establishing a faithful text-to-motion mapping. As a consequence, the CLIP encoder tends to produce text embeddings that lose semantics, while the diffusion model simultaneously adapts to interpret these distorted embeddings. This leads to text embeddings that overfit to the training distribution and fail to generalize to unseen descriptions, ultimately harming generation quality.
> We have added this discussion to Section 4.4 in the revised version.
>
> ```
> Q8: In the appendix, I find it strange why the results of AdaQF trained without using KIT-ML are better than those trained on KIT-ML.
> ```
>
> We clarify that **FID is a standard metric in which lower values indicate better performance** . In Figure 8 of the revised submission, the blue bar denotes using the vanilla CLIP, the purple bar denotes using AdaCLIP trained on HumanML3D, and the yellow bar denotes using AdaCLIP trained on KIT-ML. Among all models, using AdaCLIP from HumanML3D achieves the best FID, followed by AdaCLIP from KIT-ML.

---

### Official Review · Reviewer_mpzx · 2025-11-01

**Soundness:** 3
**Presentation:** 3
**Contribution:** 3
**Rating:** 6
**Confidence:** 4

**Summary:**

This paper introduces an innovative framework for human motion generative model, which mainly focuses on text embedding. The proposed method achieves noticeable improvement while significantly reduces the time cost of covergence.

**Strengths:**

1. The paper demonstrates impressive quantitative and qualitative results, establishing a new state-of-the-art with a substantial margin.

2. The proposed framework is novel. The associated conclusions and experiments make significant contributions to the research community.

3. The paper is well-written, ensuring that its content is easily understandable for readers.

4. The authors show the improvement brought by the proposed methods on serveral different backbones, which strongly shows the generalizability and effectiveness of the proposed method.

**Weaknesses:**

1. In the supplementary material, please include a user study as a complement to the qualitative comparison to further demonstrate the effectiveness of the improvements.

2. I have a concern that, when training the contrastive-learning model, the authors used the same architecture as the standard evaluator. This risks 'hacking' the evaluation protocol rather than genuinely improving performance. It would be better to train several evaluators with different architectures and report the corresponding results to show the gains come from the proposed training paradigm rather than a metric-specific hack.

3. It would be better if the authors report the number of parameters and inference speed of the proposed method.

**Questions:**

Please kindly refer to the weaknesses mentioned above.

---

> ### Author Response · Authors · 2025-11-20
> **Response to Reviewer mpzx (1/2)**
>
> Thank you for the constructive and helpful feedback. We respond below to your questions and comments:
>
> ```
> Q1: User Study.
> ```
>
> We sincerely thank the reviewer for this suggestion. We conducted a user study on the realism and semantic alignment of our method, MDM, MoMask, and SALAD. We randomly sampled 20 texts to generate motions, and 25 participants were asked to rank videos generated by the four methods based on realism and semantic alignment. Our method clearly outperforms both of them in terms of realism and semantic alignment. We have added the figure of the user study to Figure 11 in the revised version.
>
> ```
> Q2: Question on architecture overlap between extractors and evaluators.
> ```
>
> We thank the reviewer for raising this concern. Although extractors share the same architecture as the evaluators, they are trained with different objectives for different purposes. We clarify that our method does not "hack" the evaluation protocol. The reasons are as follows:
>
> **1. Essential difference between official evaluator and our extractors.**
>
> The official evaluator is trained using Euclidean-distance-based loss functions, enabling it to map  all the motions into a compact embedding space. This is reflected by the FID score between two set of randomly truncated ground truth motion data (GT FID = 0.002), showing that it is well-suited for FID evaluation. While it is qualified for evaluating FID, the GT R-Precision Top3 is 79.7%, which might not sufficiently represent the alignment between texts and motions in ground truth data.
>
> In contrast, our extractors are trained with cosine-similarity-based InfoNCE losses. This training objective emphasizes directional similarity between text and motion embeddings without imposing magnitude constraints, thereby preserving modality-specific information.  As a result, although our extractors excel at cross-modal retrieval (GT R@3 = 90.2%, Table 8), they are not designed for FID evaluation and naturally yield a much higher GT FID of 0.186.  In addition, R-Precision is also calculated based on the Euclidean distance, which differs from the cosine similarity we used.
>
> **2. Using official evaluators performs worse.**
>
> If our method were tailored for the evaluation protocol, then training with the official evaluator as extractors should get better results. We conduct additional experiments using the official evaluator for training, which yield the opposite results (FID: ours 0.039 vs official evaluator 0.103; Appendix Table 7). This demonstrates that the performance improvements come from our extractors and consistency constraints rather than any metric-specific bias.
>
> **3. Motivation for training our extractors**
>
> The official evaluator, trained under Euclidean-distance-based classical contrastive learning, is not optimal as an extractor during training. First, negative pairs are usually sampled randomly within a batch, which prevents the model from focusing on the other negative pairs at the same time. Second, the Euclidean-distance-based loss closely aligns text and motion embeddings in terms of absolute distance. Because text and motion embeddings are inherently heterogeneous modalities and the norm of the embeddings is modality-specific information, we believe that applying a Euclidean-distance-based constraint may cause the text and motion embeddings to deviate from their respective semantic spaces. A magnitude-agnostic cosine similarity is a better choice. Hence, we propose using InfoNCE loss (which encourages the network to attend to all negative samples) and the cosine similarity loss to train the extractors. Therefore, our method shares a totally different target from the evaluation protocol.
>
> **4. Physics-based metric.**
>
> In our ablation Table 4, we observe substantial improvements in the physics-based metric Skating Ratio. The Skating Ratio is a low-level metric that represents the proportion of frames in which foot skating occurs, which is computed independently of the evaluator. This demonstrates that our framework indeed improves the quality and physical plausibility.
>
> We hope this clarification helps. Thank you for your time and attention.

---

> ### Author Response · Authors · 2025-11-20
> **Response to Reviewer mpzx (2/2)**
>
> ```
> Q3: Additional experiments to show that the gains genuinely come from the proposed training paradigm.
> ```
>
> We appreciate the reviewer’s suggestion to evaluate with multiple evaluator architectures. Training and validating an evaluator with an entirely new architecture is non-trivial. Still, we conducted another feasible validation: we replaced our extractors with the official evaluator while keeping all training and evaluation protocols unchanged.
>
> If our method exploits metric-specific biases, we would expect it to perform perfectly when both training and evaluation use the same official evaluator. However, we observe the opposite: using the official evaluator as an extractor yields worse R-Precision (0.871 vs 0.888) and substantially worse FID (0.103 vs 0.039) than our InfoNCE + cosine-similarity-based extractors. We report the results below and have added to APPENDIX Table 7 in the revised version.  This observation aligns with our analysis that cosine similarity is more appropriate for heterogeneous cross-modal embeddings, indicating that our improvements do not come from "hacking" but from genuinely better semantic alignment.
>
> We hope this additional experiment addresses the reviewer’s concern and clarifies that our gains do not exploit metric-specific bias.
>
> | Method | R@Top1 $\uparrow$ | R@Top2 $\uparrow$ | R@Top3 $\uparrow$ | FID $\downarrow$ | MultiModal Dist $\downarrow$
> | :--- | :---: | ---: | :---: | :---: | :---:
> | with our extractors | $0.627^{±0.004}$ | $0.811^{±.003}$ | $0.888^{±.002}$ | $0.039^{±.001}$ | $2.382^{±.010}$
> | with official evaluators | $0.611^{±.003}$ | $0.803^{±.002}$ | $0.871^{±.001}$ | $0.103^{±.003}$ | $2.407^{±.011}$ |
>
> ```
> Q4: Report the number of parameters and inference speed.
> ```
> We provide comparisons of parameters and inference speed for MDM-1000steps, MoMask, SALAD, and our method. For MDM, we provide the time cost with training and sampling steps of 1000. AITS means Average Inference Time per Sample, proposed by MLD[1]. AITS calculates the average time cost of generating a sample with a batch size of one.
> | Method | Parameters  | AITS | FID
> | :--- | :---: | :---: | :---: |
> | MDM-1000steps | 177M | 13s  | 0.489
> | MoMask | 347M | 0.14s | 0.045
> | SALAD |  161M |  0.50s | 0.076
> | Ours | 177M | 0.65s | 0.039
>
> [1] Executing your Commands via Motion Diffusion in Latent Space. In ICCV2023

---

### Author Response · Authors · 2025-11-27
**General response to all reviewers and AC**

Dear reviewers and AC,
We sincerely appreciate your valuable time and effort spent reviewing our manuscript. We are glad to see that our work is recognized as **"our work is novel"** (mpzx) ,
**"impressive quantitative and qualitative result"**  (mpzx, 3KCs),
**"improvement is substantial and significant"** (mpzx, 3KCs),
**"simple yet effective"**  (3KCs),
**"faster convergence than baseline"** (3KCs),
**"make significant contributions and impacts to the research community"** (mpzx, 3KCs),
**"bring improvement to several different backbones"** (mpzx, 3KCs),
**"easy to follow and make a lot of sense"** (3KCs),
**"experiments are comprehensive"** (FR6P)
and **"structure of the paper is clear"** (FR6P) .

The changes to the paper following the reviewers' suggestions are summarized as follows:

1. Added discussion and advantages compared with LAMP in APPENDIX A.6

2. Added quantitative comparisons with several state-of-the-art methods in Tables 1 and 2.

3. Added explanation of why finetuning CLIP without the proposed consistency constraints causes a slight degradation in Section 4.4.

4. Added the discussions of using LoRA-based finetuning or full-parameter finetuning in Section 4.4, provided results in Table 4, and
provided qualitative comparisons in Figure 10.

5. Added the discussions of the architecture overlap between extractors and evaluators in APPENDIX A.3.

6. Added the retrieval performance of our proposed extractors to prove the effectiveness of our design in APPENDIX A.4.

7. Added user study in APPENDIX A.8.

8. Added more visual images in APPENDIX A.7.

The latest revision is notably more thorough, thanks to your valuable feedback!
We hope that our responses help answer the questions about our work. Please let us know if there is any additional information we can provide or clarify.

---

### Meta-Review · Area_Chair_mHAt · 2026-01-04

**Summary:**

Reviewers found the paper clearly written and empirically strong on the reported benchmarks, with indications of faster convergence and transferability. However, two main concerns were raised: 1) A central concern is that the technical advance appears close to existing language–motion pretraining or adaptation directions (i.e., LAMP). And the resulting performance is reported as very similar in several settings. It is hard to get convinced that this framing reflects a substantially new principle beyond a particular training design choice (pre-trained text encoder vs. joint trained). 2) The evaluation protocol “hacking” risk is not fully resolved: the authors argue their extractors differ from the official evaluator (cosine similarity vs. Euclidean distance), but the evidence is insufficient to fully rule out protocol-specific gains.

**Reviewer Concerns:**

Addressed: 1) LoRA vs full fine-tuning: the authors added an experiment showing full fine-tuning performs worse than LoRA-based adaptation (e.g., FID 0.073 vs 0.039), supporting the design choice. 2) Tuning the CLIP without the proposed two consistency constraints degrades the performance: the authors explained this as an overfitting or shortcut learning issue. 3) Some information about parameter count, inference time, and user study was added.

Not fully addressed: 1) the technical contributions as compared with LAMP; 2) the evaluation protocol hacking issue; 3) some minor issues: Reviewer FR6P questioned about the results in Figure 8. It is possible that there is some misunderstanding, given that the explanation in the text is a bit vague. Specifically, in the beginning of A.2, the authors have said "Having first trained AdaCLIP on the HumanML3D dataset, we substitute it for the vanilla CLIP encoder and keep it frozen when training several models on KIT-ML.", but Figure 8 shows "the model with AdaCLIP trained on HumanML3D" and "the model with AdaCLIP trained on KIT-ML". It brings up the question and confusion of whether the second one was first trained on HumanML3D and then tuned on KIT-ML.

**Reviewer Scores:**

Reviewers didn't show any indication of changing the scores. Given the unresolved technical limitations, it is unlikely that reviewers will raise their scores further.

---

### Decision · Program_Chairs · 2026-01-26

Reject